green chemistry/nanotechnology

Green synthesis, organic compounds, biosynthesis, nanomaterials, nanocomposites, environmental hazards

**Author for correspondence:**
Boris I. Kharisov
e-mail: bkhariss@hotmail.com

This article has been edited by the Royal Society of Chemistry, including the commissioning, peer review process and editorial aspects up to the point of acceptance.

# Greener synthesis of chemical compounds and materials

Oxana V. Kharissova[1], Boris I. Kharisov[2], César Máximo Oliva González[2], Yolanda Peña Méndez[2] and Israel López[2,3]

[1]Facultad de Ciencias Físico-Matemáticas, and [2]Facultad de Ciencias Químicas, Laboratorio de Materiales I, Universidad Autónoma de Nuevo León, UANL, Avenida Universidad, Ciudad Universitaria, 66455 San Nicolás de los Garza, Nuevo León, Mexico
[3]Centro de Investigación en Biotecnología y Nanotecnología (CIBYN), Laboratorio de Nanociencias y Nanotecnología, Universidad Autónoma de Nuevo León, UANL, Autopista al Aeropuerto Internacional Mariano Escobedo Km. 10, Parque de Investigación e Innovación Tecnológica (PIIT), 66629 Apodaca, Nuevo León, Mexico

OVK, 0000-0001-5214-8400; BIK, 0000-0001-7082-9357;
CMOG, 0000-0003-2877-7965; YPM, 0000-0003-3706-1504;
IL, 0000-0002-0957-3062

Modern trends in the greener synthesis and fabrication of inorganic, organic and coordination compounds, materials, nanomaterials, hybrids and nanocomposites are discussed. Green chemistry deals with synthesis procedures according to its classic 12 principles, contributing to the sustainability of chemical processes, energy savings, lesser toxicity of reagents and final products, lesser damage to the environment and human health, decreasing the risk of global overheating, and more rational use of natural resources and agricultural wastes. Greener techniques have been applied to synthesize both well-known chemical compounds by more sustainable routes and completely new materials. A range of nanosized materials and composites can be produced by greener routes, including nanoparticles of metals, non-metals, their oxides and salts, aerogels or quantum dots. At the same time, such classic materials as cement, ceramics, adsorbents, polymers, bioplastics and biocomposites can be improved or obtained by cleaner processes. Several non-contaminating physical methods, such as microwave heating, ultrasound-assisted and hydrothermal processes or ball milling, frequently in combination with the use of natural precursors, are of major importance in the greener synthesis, as well as solventless and biosynthesis techniques. Non-hazardous solvents including ionic liquids, use of plant extracts, fungi, yeasts, bacteria and viruses are also discussed in relation with materials fabrication. Availability, necessity and profitability of scaling up green processes are discussed.

# 1. Introduction

The term *Green chemistry*, offered to the scientific community in 1991 was designed for elimination or decrease in hazardous substances, trying to reduce the exposure of humans and environment to chemicals. Currently, several green processes are applied for water purification, energy generation and fabrication of electronics, medicines, plastics and pesticides, among many other goods. As a main idea, any chemical substance possesses hazardous properties, caused by their internal (molecular) structure, which can be changed/modified. The types of hazards are the following:

— physical hazards (flammability, explosive properties);
— toxicity (mortal, cause of cancer or other illnesses); and
— global hazards (decrease in ozone layer, change of climatic conditions, overheating, etc.).

Green chemistry deals with synthesis procedures according to its *12 key principles* (figure 1), without or at least with reduced negative impact [2] on human health and environment. These principles, to be effective, need to be applied simultaneously [3]. An absolutely *Green* synthesis does not exist: as an alternative, *Greener synthesis* is more truthful definition. These principles of green chemistry were described in the main classic book [4] in this area and Internet resources [5,6]. These principles are useful and simple [7], but do not cover all aspects of chemical processes, especially at scaling up (industrial chemical processes) and evaluation of costs. These processes can decrease negative impact on the environment, but they cannot completely prevent the interaction of hazardous materials with the environment. Following these and other green chemistry principles, it is possible to carry our energy-efficient processes to prevent the formation of wastes, provide economy of matter and energy, use safer solvents and renewable chemicals and yield less-hazard materials with low toxicity, as well as to carry out a smart catalysis with reduced number of side products or derivatives under degradable process design. To minimize risks for the environment, in the different steps of a chemical process in materials fabrication, several methods of control can be applied: use of alternative selected precursors, solvents, catalysts and reagents, change of target products, real-time process monitoring, shorter syntheses, etc. The magnitude of the so-called *E-factor* (environmental factor, mass of waste per mass of product, in kilogram; the range 25–100 correspond to the pharmaceutical synthesis and is considered as highest value) is used for evaluation of quantity of waste generated in a synthesis process. Another variable, 'atom economy', is related with the formula weight of the final desired product with the sum of formula weights of all precursor (initial reagents (in %)).

The advantages of greener chemistry processes are as follows, in particular:

— prevention of unnecessary wastes; the avoiding of unnecessary waste in organic synthesis can be reached by recyclability of most solvents, catalysts and reagents;
— economy of matter (atoms): minimization of loss of precursors and intermediate compounds during synthesis of final material;
— lower-hazard chemical reactions using little-toxic and safe chemical substances;
— low-toxic and safe separation agents and solvents (water, natural compounds as, for example, plant extracts);
— minimization of energy consumed (preferable/ideal conditions: normal pressure and room temperature);
— renewable sources (raw materials);
— minimization of reagents to avoid unnecessary wastes (lesser number of reaction steps and additional chemicals);
— use of most selective catalysts, allowing higher yields of reaction products;
— degradable reaction products, non-persisting in the environment;
— contamination prevention via permanent (real-time) analysis of reaction intermediates when possible;
— small quantities of reactants to prevent accidents (explosions, releases, fire);
— green processes are cheaper and cost-effective, frequently resulting in products with a better quality; and
— green reactions allow avoiding problems with many environmental regulations and laws.

Greener reactions are especially important in organic chemistry, where, due to a huge number of industrial processes with the use of hazardous chemicals and solvents, severe damage to the environment takes place. In organic chemistry, green techniques include reactions of C–H bond activation, fluorous, solid-supported, bio- and asymmetric catalysis and synthesis, use of water and other green solvents (in particular, ionic liquids (ILs)) or without any solvent, microwave-, ultrasound- and ultraviolet (UV)-assisted reactions, and use of flow reactors [8]. Two important aspects of green

**Figure 1.** The 12 principles of green chemistry. (*a*) Prevent waste (no. 1), (*b*) atom economy (no. 2), (*c*) avoid harmful synthetic methods (no. 3), (*d*) reduce product toxicity (no. 4), (*e*) use safe or no solvents (no. 5), (*f*) avoid high temperatures and pressures (no. 6), (*g*) use renewable resources (no. 7), (*h*) omit derivatization (no. 8), (*i*) use catalysts (no. 9), (*j*) design for degradation (no. 10), (*k*) in-process monitoring (no. 11) and (*l*) prevent accidents (no. 12). Adapted from Wiley [1].

techniques in organic chemistry are as follows: the development of catalyst-assisted reactions and correct choice of solvent. Aromatic chlorinated solvents are not recommended to be used, being volatile, toxic and destroyers of ozone layer. ILs, being non-volatile, non-aqueous and polar, can be used as alternative. If an organic reaction allows removal of products by distillation or extraction, the catalyst could remain in such a solvent; in this case, both of them (catalyst and solvent) can be recycled and re-used. Organic reactions can be also carried out in a supercritical (SC) medium of $CO_2$ without solvents (one option is under microwave irradiation (MW)); both techniques are considered as green. Toluene can substitute higher-toxic benzene in many reactions; biodegradable solvents are also a serious green alternative [9]. Green methods are also applied in medicinal chemistry and pharmaceutical industry and engineering, in particular for drug delivery purposes and battle against tropical and other diseases. In some descriptions of green synthesis, this one and biosynthesis are differentiated: the biosynthesis of compounds and materials includes the use of plant extracts, fungi, algae or microbes. In general, the green synthesis is dedicated to minimize the effect of toxic chemicals.

Green technologies are well developed for some areas (for instance, there are hundreds or more publications on the synthesis of metal nanoparticles), but are still at infancy in others (for example, biomaterials for tissue engineering). There are several books [9–11] and reviews [12–14], dedicated to particular aspects of green synthesis, and a special issue of *Nature* dedicated to green chemistry. In this review, we try to comprehend all possible fields of current research in distinct areas of greener technologies, related with organic and inorganic synthesis, preparation of materials of distinct dimensionality and sizes. As it will be shown below, practically all pure inorganic compounds (salts) are represented by nanosized particles. So, we will discuss below the green synthesis of organic and coordination compounds, macro- and nanosized objects.

# 2. General synthesis methods of green chemistry

## 2.1. Physical and chemical methods

The following physical methods are currently considered as well corresponding to the green chemistry requirements, not requiring hazardous solvents, avoiding pollution and accelerating synthesis processes: mechanosynthesis, microwave-assisted and hydro(solvo)thermal reactions (as well as their

combination), ultrasound-assisted processes, UV-irradiation of the reaction system, among others. Some of these methods belong to less-common techniques, as, for example, magnetic field-assisted synthesis. Main recent reviews in these fields are described in [15–20].

### 2.1.1. Ball milling

Chemical reactions can be efficiently driven by mechanical energy via ball milling [21], belonging to the mechanochemical (tribochemical) synthesis, widely used in organic chemistry, in addition to classic applications in inorganic solid-state processes [22]. As an example of equipment, Retsch planetary-type high-energy ball mill is described in [23]. The use of ball milling allows us to increase energy efficiency and, at the same time, avoid toxic reagents and solvents. These reactions, as well as MW-assisted processes, take place without the use of solvents and at room temperature. Being simple and environmentally friendly, the ball milling is also considered as a green tool for chemistry, but it is not used in a widespread manner by chemists, despite its big potential. Among other reactions, carried out by ball milling, are those with the use of solid oxidants and reductants for oxidation and reduction purposes, respectively, dehydrogenative coupling, synthesis of polymers, amino acids and peptides, coordination compounds, composites 'cellulose-plastic', asymmetric organic reactions using catalysts. For particular organic chemistry reactions, where organic substrates are sensitive to temperature conditions, the mechanosynthesis can be essentially useful and is currently applied, among others, in the reactions of the formation of C–C bonds (i.e. Suzuki and Glaser coupling, Knoevenagel condensation, McLurry, Wittig, Gewald, Michael, Reformatsky and Gringard reactions, arylaminomethylation, etc.), C–N bonds (synthesis of oximes, imines, azines, guanidines, (thio)semicarbazones, nitrones, N-arylation of amines, etc.), C–O, C–S, C-Hal, C–H and other bonds, cycloaddition reactions, reductions and oxidations, as well as for functionalization of nanocarbons (single-walled carbon nanotubes (SWCNTs), graphene and $C_{60}$) and in supramolecular chemistry.

### 2.1.2. Microwave irradiation

This is a green source of heating both in organic and inorganic synthesis, based on the conduction and dipolar polarization. A series of organic molecules have been reported as obtained this economic, efficient, fast and clean way, now recognized as a conventional synthetic chemistry tool, which has made a great contribution to organic synthesis [24]. MW energy is *non-ionizing radiation*, which does not influence on the molecular structure of compounds. The MW coupling of a substance depends on its *dielectric constant*, so *N,N*-dimethylformamide (DMF), methanol, acetone, water is rapidly heated under MW irradiation, in a difference from $CCl_4$, toluene or aliphatic hydrocarbons. The electromagnetic energy, being transformed into heat, drives interactions between compounds. MW irradiation and reaction components are in a *direct* interaction, so a minimum of energy is needed for its heating, without expanding the process to the furnace material [25]. That is why the temperature profiles of conventially and MW-heated simples are different (in the MW case, the interior is hotter and the surface is cooler). The heating is *uniform* and takes place *volumetrically*, in all the sample; the heat transfer requires lesser energy due to thermal conduction inside the sample. The MW heating is almost *instantaneous*, very fast, due to the fast transfer 'MW energy—heat', and can be stopped immediately by a simple turn-off of the MW equipment. In addition, the MW heating is *selective*, since reagents possess distinct capacity to be MW-heated, allowing, for example, the heating active points in a sample instead of heating an entire sample.

MW heating has the following *advantages*, among others, being compared with conventional heating: fast process speed, pure products, lesser heat loss, high heating efficiency, less waste, low cost of operations, lesser possibility of side products. *Solvent-free* (dry media) conditions of MW-assisted reactions are preferable, to avoid fast uncontrolled solvent heating and violent explosions. Precursors can be adsorbed on *inorganic supports*, transparent for microwaves (montmorillonite clays, zeolites, ceramics, $Al_2O_3$, $SiO_2$) or, on the contrary, possessing strong MW-absorbance (for example, graphite). These supports can also contain additional reagents or catalysts. MW-assisted solvent-free reactions on inorganic supports at relatively low bulk temperature have shown certain advantages in the processes of reduction and oxidation, deprotection and protection, condensation, rearrangement, heterocyclic synthesis, etc. leading to important products such as HCN, imines, nitroalkenes, enamines, among others.

Modern apparatus for MW heating is well developed and covered in a comprehensive review [24]; a commercial example is a reactor Monowave 50 [26]. For simple experiments, modified domestic MW ovens are frequently used. Commercial equipment includes SMC (apparatus with single-mode cavity,

0.2–80 ml volume of sealed vessels), CEM MARS multimode MW reactor and Milestone Flowsynth continuous-flow reactor, all with maximum pressure of 300 psi and temperature 300°C. Sometimes, ultrasmall particles (5–10 nm) can be formed due to very rapid reaction time, although in some cases, *larger* particles can be also formed in comparison with conventional synthesis. Crystallinity, defect concentration and morphology can be different in MW-assisted and conventional synthesis. Shapes of products can depend on MW power and irradiation time, as, for example, in $MoS_2$ synthesis, where the layer-, nanotube- and fullerene-like forms were observed depending on the MW-treatment duration.

To carry out a successful MW-synthesis, all the reagents should ideally be *good absorbers*, otherwise, it is necessary to looking for another material acting as a heat source (susceptor) and converting MW energy to heat. The last case is not desirable at scaling up processes, since this additional material is a contamination source. Also, the substances, transparent for MW irradiation before certain temperature, can be conventionally heated to reach it and then the MW-assisted process starts. The use of MW-induced plasma, transferring energy between radiation and reagents and being, in some cases, a source of reactive species, is also possible.

Green solid- and liquid(solution)-state MW-synthesis of inorganics includes the formation of simple (in particular, yttria-stabilized $ZrO_2$, $\alpha$-$Fe_2O_3$ thin films) and double oxides (i.e. Li, Cu, Co, Ni ferrites, $BaTiO_3$ (from $TiO_2$ and $BaCO_3$), $LaCrO_3$, $La_{1-x}Sr_xCoO_3$, among others) during some minutes; chalcogenides (ZnS, CuS, $Cu_{2-x}Te$, HgTe, $Cu_2SnSe_4$, $Bi_2Se_3$, CdSe, etc. sometimes from sulfates by carbothermal reduction, but generally from elemental powders in vacuum, avoiding chalcogene sublimation, without or with susceptors); borides ($MgB_2$, $ZrB_2$); carbides (SiC, $MgNi_3C$, $CaC_2$, $Mo_2C$, WC, etc. from metals or their oxides ($Ta_2O_5 + C$)); silicides ($Li_{21}Si_5$, $MoSi_2$, $Mg_2Si$, $CaAl_{1-x}Cu_xSi$, etc.); nitrides and other pnictides (TiN, $Ta_2N$, AlN, $LiSi_2N_3$, GaN, $Li_3Bi$, $Li_3Sb$); carbonitrides $V_{1-x}Ti_x$(C, N), and oxynitrides ($\beta'$-SiAlON).

### 2.1.3. Photocatalysis

Photochemical reactions under UV-irradiation are considered as green chemistry interactions and are based on the electronic excitation, which influences the chemical reactivity of reagents in organic synthesis. A recent review on photocatalysis [27] describes generation of singlet oxygen and its role in the photo-oxygenation (incorporation of molecular oxygen into molecules), combination of photochemical processes with enzyme catalysis, application of continuous flows or microreactors for their optimization. Some examples of such reactions are the synthesis of N-containing heterocycles by photo-oxidation of furan derivatives, asymmetric oxidations catalysed with enzymes, and preparation of several F-organic compounds by photocatalysed trifluoromethylation of aromatics, using photocatalysis booth and reactors described in [28].

### 2.1.4. Hydro(solvo)thermal synthesis

A solution reaction-based approach [29] is applied to synthesize compounds, crystallize [30] and grow single crystals and polycrystals at high pressures (normally up to 10 bar) and elevated temperatures (generally up to 300°C) in water or organic solvent media. The equipment for hydro(solvo)thermal technique consists of an autoclave (thick-walled steel cylinders, sometimes with protective inserts made of Teflon, platinum, titanium, quartz, gold, etc.), containing solvent (water) and dissolved/suspended precursors. For the past 20 years, a combined equipment 'microwave-hydrothermal treatment' has been frequently used in laboratories. The main advantage of the hydrothermal method is the possibility of the formation of crystalline phases, unstable at the melting point of the desired compound; the main disadvantage is the necessity to have expensive equipment. Morphology control and crystallinity for the formed materials can be made by changing pressure, temperature, solvent, reaction time or precursors' ratio. Hydrothermal reactions in water are considered as more suitable for green chemistry purposes, being environmentally friendly, and are widely applied to fabricate a variety of materials. This method allows minimum loss of reactants and frequently higher yields of products, being especially useful to obtain classic and less-common nanostructures with desired shape and size control: powders, films and especially one- to three-dimensional nanocrystals [31,32].

### 2.1.5. Ultrasound-assisted (sonochemical) synthesis

Ultrasound-assisted (sonochemical) synthesis is only a solution-based approach (since cavitation occurs only in liquids) working as a result of the phenomenon of the acoustic cavitation (bubbles, appearing in a

liquid, grow and implosively collapse, leading to extreme local pressures of 1000 atm and temperatures of 5000 K) in a liquid phase, without a direct affectation to the bond vibrational energy. Chemical reactions can be started or intensified in these conditions, additionally to the formation of free radicals and $H_2O_2$. Chemical composition, reactivity and surface morphology of materials (frequently increasing surface area) can be considerably changed as a result of these collisions. A variety of sonochemical reactions can be carried out in liquid systems, solid–liquid and liquid–liquid interfaces (as well as sonocatalysis), useful for the environment, for instance, destruction of halogenated aromatics. Ultrasound-mediated organic synthesis [33] and preparation of functional materials [34] are classic contributions, perfectly fitting [35] to the green chemistry field, since they do not require large energy consumption and hazardous chemicals; additionally, the cost of equipment is normally low in the case of a simple ultrasonic cleaning bath (frequencies 20–40 kHz), with exception of high-power ultrasonic horns [36].

## 2.1.6. Magnetic field-assisted synthesis

Magnetic field-assisted synthesis is currently studied as an alternative to traditional methods. because some of the traditional methods require the use of toxic solvents or additional steps that need more energy and can generate unwanted residues. These extra steps can be omitted using magnetic fields during the synthesis of the desired material. The synthesis assisted by magnetic fields allows obtaining morphologies different from those prepared by traditional methods. In addition, it is possible to influence the orientations of the final product planes. Some synthesis techniques where the assistance of magnetic fields can be used are the methods of coprecipitation, solvothermal, electrospinning, etc. In addition, the magnetic field assistance can be implemented directly in the solution to influence the growth of the crystals of the desired material and can be applied for the formation of thin layers of some composites. The synthesis assisted with magnetic fields has numerous advantages due to its simplicity; however, it has some important limitations, derived from the foundation of the technique, because it can only be applied to materials whose reactants or products have magnetic properties; that is why the vast majority of reports in the literature use magnetic iron oxides.

Iron oxides such as $Fe_3O_4$ can adopt different assisted morphologies with magnetic field; for example, it is possible to obtain complex $Fe_3O_4$ nanorods by the solvothermal technique. This is possible because, during the formation of nanorods, the growth of the crystals of the metal oxide is oriented in a specific orientation due to the magnetic field [37]. If the magnetic field is applied in the $Fe_3O_4$ synthesis by thermal decomposition, it is possible to obtain nanotubes of 250 nm, which are not formed in the absence of the magnetic field; besides, it was observed that the speed in which these nanoparticles of $Fe_3O_4$ are heated by hyperthermia is considerably higher in the oxides synthesized in the presence of the magnetic field [38]. Another example that can be mentioned of the synthesis of $Fe_3O_4$ with the assistance of magnetic fields is a series of chains prepared at a temperature of 90°C. These particles had an average size of 150 nm. This was achieved due to the growth of the particles of uniaxial way by the influence of the magnetic field, and it was observed that the intensity of the magnetic field during the synthesis had a significant effect on the anisotropy of the resulting material [39].

The assisted syntheses with magnetic fields can also be applied to the synthesis of some composites, as long as they comply with the characteristics mentioned above. An example of this is the thin layers of reduced graphene oxide (rGO) with $Fe_3O_4$ nanocrystals. The synthesis of this composite is quite simple, based on taking advantage of the ferromagnetic characteristics of the nanocrystals of $Fe_3O_4$ that are adhered to the surface of sheets of rGO. When applying the magnetic field on this suspension, the particles agglomerate in one of the walls of the reactor and they adhere to each other by the electrostatic interactions between the GO and the nanocrystals of $Fe_3O_4$ [40]. Such techniques can support the synthesis of flexible electrodes for batteries of new generation.

Some other materials reported to assist with magnetic fields are compounds containing nickel, cobalt, bismuth and some of their alloys [41–43]. This is due to the fact that, as mentioned, it is necessary that the materials have ferromagnetic properties, this being their only limitation. Although it is true that there are reports of iron oxides in the literature, it is also possible to find reports, where assisted synthesis with electromagnetic field has been made using other materials and compounds containing nickel, cobalt, bismuth and some of their alloys with ferromagnetic properties [41–43]. For example, the synthesis of cobalt and cobalt carbide nanowires is known, using magnetic field-assisted polyol synthesis. For comparison, both cobalt and cobalt carbide synthesized by traditional methods in the absence of magnetic fields usually have spherical morphologies [44]. It is also possible to obtain thin films of cobalt nanosheets using magnetic fields during the reduction in solution of the

Co$^{2+}$–EDTA complex, which opens a new methodology to create thin films of magnetic nanocrystals with anisotropic forms [41].

The synthesis of bismuth by the solvothermal method assisted with a magnetic field can also be used to generate nanowires [45]. As it can be seen, the use of magnetic fields has a particular influence on the orientation in the case of the formation of crystals of ferromagnetic materials and in the order in which the composites assisted with magnetic fields are accommodated. The use of this technique depends on the method that is being complemented to be considered as a green synthesis. It is noteworthy that this technique reduces waste and synthesis, helping to make it more efficient; in addition, the application of the magnetic field does not generate any additional by-product to the main technique used.

## 2.1.7. Solvents and catalysis in green processes

For the organic synthesis, where hazardous solvents are mostly used, the 12 rules of green chemistry can be successfully applied [4,12]: maximal atom economy (avoiding by-products and wastes, in particular by solvent-less techniques, i.e. dry media), safe and non-hazardous chemical routes without harmful chemicals, use of renewable precursors (i.e. plants instead of fossil fuels), catalysts in small amounts (non-harmful and preferably solids in order to be renewable), safer chemicals and solvents (water, ILs) or better their absence, biodegradable substances, as well as avoiding energy waste, preventing pollution and minimizing possibility of accidents.

*Solvents* represent a major source (80–90%) of pollution in organic (and not only) chemical processes. Conventional organic solvents, derived from oil, are toxic and possess bad health, environmental and safety impacts. So, the solvents to be used as green media for organic synthesis need to possess biodegradability, low toxicity, high boiling point, easily recyclable and non-miscible with water. Upon these conditions, the water, ILs, polyethylene glycols (PEGs) and some SC fluids are most appropriate. In this respect, non-toxic and biodegradable glycerol, which is the principal side product in the fabrication of biodiesel, is quite suitable for green synthetic organic synthesis and even called 'organic water'. Its advantages are great availability, biodegradability, low cost, low vapour pressure, high boiling point; in addition, glycerol is immiscible with aliphatic hydrocarbons, highly polar, capable to form hydrogen bonds and solubilizes both inorganic and organic compounds. The review [46] describes a host of reaction schemes (metal-free, metal- and biocatalysed transformations) with glycerol use.

*Supercritical green solvents*. Among other greener solvents, non-flammable, non-toxic and environmentally friendly SC CO$_2$ is known from long ago as a good alternative solvent for the synthesis of polymers [47]. SC water has been also used as a green solvent [48]. Currently, the technologies based on the supercritical fluids (SCFs), in particular SC water or CO$_2$, contribute to the green chemistry in the nanomaterials synthesis (for example, in semiconductors' fabrication using SC CO$_2$), solving some environmental problems. The SCF technologies have been used in materials synthesis processes such as extraction, cleaning, fractionation, drying, polymerization, hydrothermal reactions, plating, biomass conversion, dyeing, among others, providing solvent-free media, simplicity and recyclability, high yields, absence of wastewater and secondary pollution, etc. [49].

In the case of SC water (above critical pressure 22.1 MPa and critical temperature 646 K), it can be mixed with organic and inorganic substances forming homogeneous phase. The reactions in SC H$_2$O can occur by different routes due to the changes in thermal conductivity, diffusion coefficient, density of molecules and viscosity. Additionally, the dielectric constant of water (78 at room temperature (r.t.)) decreases to values typical for polar organic solvents around the critical point. Metal oxides can be hydrothermally produced in SC water, sometimes with H$_2$O$_2$ addition to create oxidizing atmosphere. A considerable decrease in their solubility near the critical point was observed and considered as a good advantage. Resulted green reactions have led to the fabrication of a variety of metal oxides, suitable for applications in the catalysis (for example, nanoparticles of TiO$_2$), as sensors, coatings, semiconductors, devices for solar energy and batteries (in that number, LiMn$_2$O$_4$, LiFePO$_4$ and LiCoO$_2$), ceramics, as well as for distinct medical applications. These methods are indeed green due to the use of water and especially because they lead to the fabrication of lower-cost sustainable nanomaterials. Metal oxide nanoparticles, modified with organic functionalities, can be also obtained in these conditions, providing good dispersibility in water or organic solvents. Other examples of SC-hydrothermally obtained materials are hybrid materials based on inorganic nanoparticles and polymers, possessing properties of both counterparts (high electrical resistance, mechanical strength, and thermal conductivity, light weight, etc.). Also, recycling materials (plastics) can also be made in SC water and EtOH; in these conditions, their decomposition or depolymerization occurs to form

monomers. Recovery of materials from wastes is also available, in particular by hydrolysis of plastics in SC water or by SC oxidation of wastewater.

ILs, some of the best solvents after water for green chemistry goals, are indeed salts (ionic compounds, 'constructed' of ions and short-lived ion pairs), which are in a liquid state at 100°C and below (usually at room temperature). In their composition, mostly used cations are $N$-alkylpyridinium, 1-alkyl-3-methylimidazoium, $PR_4^+$, $NR_4^+$, typical anions are $[PF_6]^-$, $[BF_4^-]$, $[CF_3SO_3^-]$, tosylate, alkylsulfate, $[CH_3COO^-]$, $NO_3^-$, $Cl^-$, etc., common alkyl chains are ethyl, butyl, hexyl, octyl and decyl. IL molecules have a significantly lower symmetry in comparison with classic ionic compounds. In a difference with water, where hydrogen bonds act between molecules, and organic solvents with van der Waals interaction, in ILs, the Coulomb interactions are main forces. These solvents are generally thermally stable, electrically conducting fluids, highly viscous and highly powerful solvents with low vapour pressure (approx. $10^{-10}$ Pa at 25°C); some of them are magnetic, for instance 1-butyl-3-methylimidazolium tetrachloroferrate. The solubility of chemical compounds in ILs depends on the ability of hydrogen bonding and polarity. In addition to a solvent's role in chemical processes, ILs are used for gas handling, coal processing, fabrication of pharmaceuticals, nuclear fuel reprocessing and cellulose processing, among several other uses. Currently, about 1000 ILs are known, including 300 commercially available products.

The *catalysts* in organic synthesis are used in the homogeneous and heterogeneous catalysis, acting in the same (normally in liquid phase) or different phase (typically solid/gas, liquid/gas or solid/liquid) in the reaction mixture, respectively. In the first case, the active sites of catalysts are separated from each other. In the second case, phase separation 'product-catalyst' is easier, since no extraction or distillation is needed, thus contributing to the green chemistry. In addition, solid hazardous catalysts (for example, polluting and corrosive catalysts) can be substituted by greener zeolites or clays. *Catalyst-free* organic synthesis as the best greener alternative occurs for several reasons, limiting catalyst use, for instance, lower selectivity, longer synthesis duration, higher catalyst amounts, toxicity, insolubility, high cost, etc.

*Dry media* in organic synthesis is obviously a great contribution to the green chemistry in the conditions of solvent-free interaction of reagents, which can react directly or be incorporated in alumina, silica, zeolites or clays. Physical treatments described above (heating, UV-irradiation, ultrasound or microwaves) can be also used in these reactions. The processes are shorter, with reduced contaminations, economic and frequently scalable.

As examples of replacing dangerous procedures with greener methods [50], we note, for example, use of $Na_2CO_3$ and Zn instead of fusing metallic Na with organic substances for organic qualitative analysis, use of LiOH instead of NaOH in the synthesis of dibenzalpropanone from acetone, benzaldehyde and EtOH, solvent(EtOH)-free preparation of benzylic acid, bromination of *trans*-stilbene in EtOH instead of chlorinated solvents, etc.

## 2.2. Biological methods

Biological methods, being compared with conventional chemical and physical methods, could be a preferable synthesis route due to environmentally friendly conditions, despite their lower speed of metal reduction. However, their studies are currently relatively limited, especially those establishing key factors of the biosynthesis, so this method is only developed in laboratory scale, although bacteria could be applied for industrial recovery of silver. The biological synthesis of nanoparticles does not require further stabilizing agents: the microorganisms or extracts themselves act as stabilizing and capping agents. Other advantages of the biological routes are no necessity of toxic chemicals and contaminants, possibility to control shapes and sizes, low cost, biocompatibility and numerous precursors (microorganisms and plants). So, biological methods perfectly fit to the green chemistry, in particular to nanochemistry, resulting in biologically produced nanoparticles, which are non-toxic, stable, environmentally friendly and cost effective. Frequently, the nanoparticles, prepared by biological methods, expose a higher antimicrobial activity (being compared with those synthesized by conventional chemical methods), related with the fact that the stabilization and capping by proteins is more effective. Main recent reviews in this area are described in [51,52].

Biology-based green chemistry methods consist of the use of *bacteria, viruses, yeasts, plant extracts, fungi* and *algae*, among which we consider plant extracts as most frequent and popular green routes [53], as it will be shown below especially for the synthesis of nanoparticles, not only those of noble metals, but also carbon dots, metal sulfides, oxides, etc. This area, phytonanotechnology (use of plants) is scalable and medically applicable. Microorganisms represent natural nanofactories, capable to adsorb, accumulate and reduce toxicity of heavy metals, where enzymes are able to reduce metal ions to cero-valent nanoparticles. In the case of bacteria use for nanoparticle preparation, the

techniques include the use of bacteria-containing biomass, as well as supernatant and derived components. Mycosynthesis (use of fungi) allows an easy, stable and possibly scaled-up biosynthesis of nanoparticles. Viruses, whose feature is dense and highly reactive surface (for instance, the surface of *tobacco mosaic* virus contains 2130 capsid protein molecules), were used to produce several inorganic nanoparticles, such as CdS, ZnS, $Fe_2O_3$ and $SiO_2$, important for semiconductor and quantum dot (QD) applications. In addition to conventional 0D nanoparticles, the nanotubes and nanowires can also be produced by virus action. The use of yeasts led the preparation of CdS and PbS QDs, Au and other metal nanoparticles. The use of algae is rare, generally for Au, Pd and Pt nanoparticles. Thus, in the case of algae use, the tetrahedral, decahedral and icosahedral Au nanoparticles, as well as bimetallic Ag/Ag nanoparticles, can be formed due to the action of proteins as stabilizing agent, shape-control modifier and reductant at the same time.

Inorganic micro- and nanosized materials can be prepared via *intracellular-* and *extracellular-* microorganism-assisted synthesis. The formation of nanoparticle is considered (but not yet fully understood) as a bottom-up process due to redox processes with participation of metal ions and biological molecules (proteins, sugars and enzymes), provided by the microorganism, whose interaction with metal ions depends on its type and can vary, also depending on temperature and pH (environmental factors), as well as metal salt concentration, leading to a particular morphology and size of nanoparticles. In the intracellular methods, several additional stages are needed, such as ultrasonication to destroy cell wall, washing, centrifugation, etc. which are absent in the extracellular techniques.

Biosynthesis processes of fabrication of homogeneous-size nanoparticles with certain reproducible morphology is carried out by size and shape control of *critical factors*, such as temperature (maximum possible for fast growth of microorganisms), pH (one of the most important factors, ranged generally from 3 to 10; acidic pH contributes to the agglomeration of nanoparticles due to the higher ion binding), salt concentration, choice of biological source, temperature, redox conditions, synthesis duration and incubation period, irradiation and aeration.

As it was noted, the use of plant extracts are common preparation procedures for a variety of nanoparticles of metals, non-metals and several of their compounds. Plant extracts contain polyphenols, terpenoids, proteins, enzymes, peptides, sugars, phenolic acids, bioactive alkaloids as a driving force for nanoparticle formation. Using plant extracts, the following nanoparticles have been obtained: Au, Ag (a host of reports), Cu, CuO, $TiO_2$, ZnO, $In_2O_3$, Fe, $Fe_2O_3$, Pb and Se. A large number of their final morphologies have been reported: spheres, triangles, cubes, pentagons, hexagons, wires, rods, etc., for example, Au triangles (reduction of $HAuCl_4$ with *Aloe vera*), Ag nanowires (reduction of $Ag^+$ with *Cassia fistula* leaf). Nanoparticle sizes can vary considerably, for example, CdS (from ultrasmall 2–5 nm to large 200 nm), Ag (5–400 nm), Au (5–85 nm) or magnetite (20–50 nm). Factors influencing their size and shape are as follows: plant extract and metal salt concentrations, temperature (25–95°C), pH (lower pH—larger particles), reaction time (normally from minutes to several hours).

Different metals have distinct capacity to be reduced by plant extracts; easiest processes correspond to noble metals, especially silver. Typical disadvantages in plant-assisted reduction of metals are difficulties in separation of formed metal nanoparticles from the biomass and accompanying putrification processes in reaction systems in the case of long-term reduction. So, the low or practically no cost of plant extracts can cause an imagination about apparent scalability for these processes; however, necessity of additional separation steps can result a negative conclusion on potential pilot-plant or major fabrication of metal nanoparticles.

A huge number of reports are dedicated to $Ag^0$, $Au^0$ monometallic and $Ag^0/Au^0$ bimetallic nanoparticles, obtained by biological methods and reviewed in [54–57]. Silver nanoparticles have been biologically obtained by all possible methods [58], including bacteria-assisted bioreduction (in particular, using *Escherichia coli*, *Bacillus licheniformis*, *B. subtilis*, among others), fungi (*Aspergillus fumigatus*, *Fusarium acumina-tum*, *Penicillium fellu-tanum*, among others), yeast (yeast strain) and numerous plant extracts containing polyphenols as reductants (pennyroyal—*Mentha pulegium* L., *Platanus orientalis*, thyme—*Thymus vulgaris* L. [59], *carum carvi* L. seeds [60,61], black and green tea, lemon grass, grape wastes, orange wastes, *Magnolia kobus*, marshmallow flower (*Althaea officinalis* L.), geranium, etc.). Au and Ag nanoparticles can be obtained with the use of actinomycetes (microorganisms with some properties of bacteria and fungi) and leaves of *Passiflora edulis* [62,63]. Sometimes additional microwave treatment and capping agents were applied to complete metal formation process. Using microorganisms, metal ions (for instance, $Au^+$) can be reduced on cell membranes and walls in the intracellular reduction as a result of releasing enzymes from cell membrane. At the same time, the proteins can take a capping role. In the extracellular reduction, nitrate reductase enzyme can be a driving force for metal ion ($Au^+$, $Ag^+$) reduction according to the electron transfer mechanism and further stabilization by proteins.

# 3. Green synthesis of organic, labelled and hybrid compounds

A host of organic synthesis reactions, considered by authors as green processes, have been recently reviewed [1,64,65]. So, here we briefly mention the most representative recent examples of ecofriendly organic reactions, which fit to the 12 green chemistry rules. Thus, among N-containing heterocycles, certain attention has been paid to benzimidazole derivatives, in particular to benzimidazole–diindolylmethane hybrid compounds [66]. Hybrid regioisomers of 2-(3,3′-diindolylmethylphenyl)-1*H*-benzimidazole were obtained without *catalysts* and *solvents* under *microwave* treatment in short time (3–8 min) and with good yields, as a result of combination of several green chemistry principles. Also, important recent small molecules, prepared via greener processes, and comparative advantages of applications of greener methods for obtaining N- and O-containing compounds are as follows:

— a series of five, six, seven and eight-membered *pyrrolo[1,2-a]benzimidazoles*, prepared [67] using hydrogen peroxide in ethyl acetate from commercial *o*-(pyrrolidin-1-yl)anilines as precursors without further steps such as chromatography or organic-aqueous extraction;
— tetra-substituted *pyrrole derivatives*, synthesized with good yields (88–93%) in one step under ultrasound-assisted standard Knoevenagel condensation in water as a solvent and without catalyst followed by the reaction of Michael type [68];
— for 1,3,5-triazine, the use of silica supported Lewis acids ($TiCl_4$, $ZnCl_2$ and $AlEt_2Cl$ supported on silica gel) under microwave conditions leads to *tris-pyrazolyl-1,3,5-triazines*, where triazine and pyrazole rings are connected with C–C bonds [69];
— synthesis of *adipic acid* from cyclohexene as precursor by oxidation with $H_2O_2$ in a microemulsion system in the presence of stearyldimethylbenzylammonium chloride without $N_2O$ emissions (as in a conventional method), extreme reaction time, pressure, mixing and temperature, applying non-dangerous ultrafiltration instead evaporation;
— synthesis of *soluble polyphenol* via a simple formaldehyde-free route by direct oxidation of phenol by potassium ferricyanide in alkaline water; compared with classic routes, the use of dimethyl carbonate as an alternative reagent under solid/liquid phase conditions at 90°C for 5 h led to final products, methyl eugenol and veratraldehyde, in 95–96% yield, avoiding toxic compounds [70];
— higher-yield synthesis of *propylene carbonate* (PC) in solvent-free conditions by exothermic cycloaddition reaction of $CO_2$ and propylene oxide in a Parr high-pressure reactor using some heterogeneous catalysts (Ce–La–Zr–O) [71]; and
— synthesis of *hydroquinone* by several steps (benzene-cumene-phenolhydroquinone) from petroleum-derived benzene as a precursor, using the Rh on $Al_2O_3$ catalyst with 59% yield [72].

Other recently developed greener techniques deal with compounds such as poly(tannin-hexamethylendiamine), which can be used as an effective adsorbent for Cr(VI) [73], a lipophilic cantharidine-like model compound for pharmacological probe [74], thalidomide and some of its derivatives [75], 3,4-dihydropyrimidin-2(1H)-ones [76], biologically important spiro-2-oxindole derivatives [77], methyl nitroacetate [78]. Milder conditions, higher yields and other advantages are characteristic features for these processes. We note that several compounds above are industrially important, for example, hydroquinone or PC. Indeed, a *variety of particular objectives* and goals can be reached by greener methods, in particular:

— use of molecular $O_2$ in C–H bond amination without solvents, metals and additives [64];
— five times reuse of the catalyst in the reaction of aryl bromides and arylboronic acids [79];
— use of water-tolerant Lewis acid $ZrO(NO_3)_2$ catalyst for cyclization of *o*-aminochalcones [70];
— use of cheap carbonyl Fe powder [71];
— high-yield short-time solvent-free reactions [72];
— bubbling with $H_2$ without autoclave [73];
— high-yield synthesis of NHPI esters in water [80];
— use of commercial montmorillonite K10 as a catalyst [81]; and
— use of cheap $H_2O_2$ as oxidant [82].

In addition to classic organic molecules, a review [1] is dedicated to the green synthesis of 'inorganic–organic hybrid materials', consisting of an organic (polymer) matrix containing inorganic building blocks (1–$10^3$ nm). In these hybrid materials, the amount of inorganic polymeric component is considerably less than that of the organic component. Generally, such materials have an enhanced

mechanical strength and a lot of other features, related with electrical conductivity, magnetism, redox or catalytic activity, and can considerably improve existing technologies for electronic devices, fuel cells, membranes, etc. The same green chemistry requirements, suitable for organic synthesis, can be satisfied in the synthesis of hybrid materials too. Thus, the organic and inorganic components of hybrid materials need to be compatible, so surfactants are often required, preferably biodegradable and biocompatible. For both components—organic and inorganic—green methods were analysed apart. In the case of inorganic colloids, they can be produced from renewable resources in some cases, additionally to be synthesized by assistance of organisms or via hydrothermal reactions [83]. In the case of polymers (organic component), they can be polymerized from non-high-molecular precursors by green routes and from renewable resources. In the case of *labelled compounds*, substitution of H hydrogen isotope with deuterium (D) in drugs reinforces the drug resistance against metabolism, increases duration of drug function, decreases side effects and necessary dosage. As a greener method for D introduction (deuteration method), Al-$H_2O$ system was offered for exchange water by $D_2O$ [84], resulting in a safe deuterium source. Additionally, ultrasound-assisted treatment can increase the activity of aluminium metal; in some cases, MW can accelerate processes of H–D interchange in amino acids and drugs, resulting final products for 1 h.

# 4. Metal salts, complexes and MOFs

In this section, we will discuss how several metal salts of organic acids, coordination compounds and metal-organic frameworks (MOFs) can be obtained by greener methods [85]. Recent reviews in this field are scarce [86–88]. In the case of using alkaline- and alkaline–earth metal complexes as final products or reactants, high-purity, highly thermally stable, crystalline and blue photoluminescent bis(8-hydroxyquinoline) calcium ($CaQ_2$) was prepared [89] with high yields from 8-hydroxyquinoline and $Ca(OH)_2$ (2 : 1 molar ratio) in water by stirring at 90°C for 4 h, being proposed for use as organic electroluminescent materials. Another example is an improved method for transformation of meseylates or glycidyl tosylates to allyl alcohol derivatives, applying excess amount of reducing agent rongalite (approx. 3 mol equivalent, sodium hydroxymethanesulfinate dihydrate $Na^+HOCH_2SO_2^-\cdot 2H_2O$) with catalytic amount of tellurium (0.1 mol equivalent) as catalyst material [90]. The purpose was to avoid difficulties in elimination of elemental tellurium, formed from telluride anion, during this conversion, resulting in excellent yields (up to 92%). This process can be considered as greener due to the recyclability of tellurium and its use in small catalytic amount, conversion of rongalite (commercially available, inexpensive and powerful reagent for organic synthesis) to bisulfite derivative, which is non-toxic and water-soluble, and use of water as principal solvent with addition of small volume of tetrahydrofuran (THF).

For *metal phthalocyanines* as important compounds with a host of applications, new sustainable synthesis methods are in a permanent search. In general, industrial production of phthalocyanines requires relatively high temperatures (above 180°C) and hazardous solvents. Phthalocyanines in a crystalline form are of especial interest. Thus, prism-shaped crystals of cobalt phthalocyanine (CoPc, more than 1 mm in size, stable up to 430°C) and zinc phthalocyanine (ZnPc, up to 8 mm in size, stable up to 550°C, figures 2 and 3) were prepared at 160°C from *o*-phthalodinitrile and metal acetates as precursors for 6 h in one-stage solvothermal process, using EtOH, pentanol or benzyl alcohol as reaction media [91]. Change of solvent led to different length of phthalocyanine needles. This greener technique does not require any additional substances, in particular surfactants, or recrystallization with the use of concentrated sulfuric acid. Since MPc single crystals are important materials for construction of devices, among other applications, the search for appropriate environmentally friendly methods for their synthesis and growth contributes to rapid development of organic semiconductor devices.

Greener synthesis of *MOFs* is also underway in several laboratories worldwide [92]. In relation to green chemistry, the crucial parameters in the MOF synthesis consist of: (i) solvent, (ii) cation source (inorganic part), (iii) linker molecules (organic part), and (iv) synthesis conditions (pressure, temperature, reactor). In the case of wastes in MOF syntheses, their big amounts are formed as a result of solvent use both in the synthesis and for purification purposes; therefore, greener solvents are needed, for instance, ethanol or acetic acid. Maximum use of all precursors should be taken into account: any waste or side product should be re-elaborated that leads to the loss of time and energy. Metal oxides or hydroxides (lesser sulfates) are recommended to be used as metal source, thus avoiding loss of toxic anions (perchlorates, nitrates or chlorides) due to water formation from O atoms of oxides and protons from acidic linkers. Solvent-less and energy-minimized methods are preferable

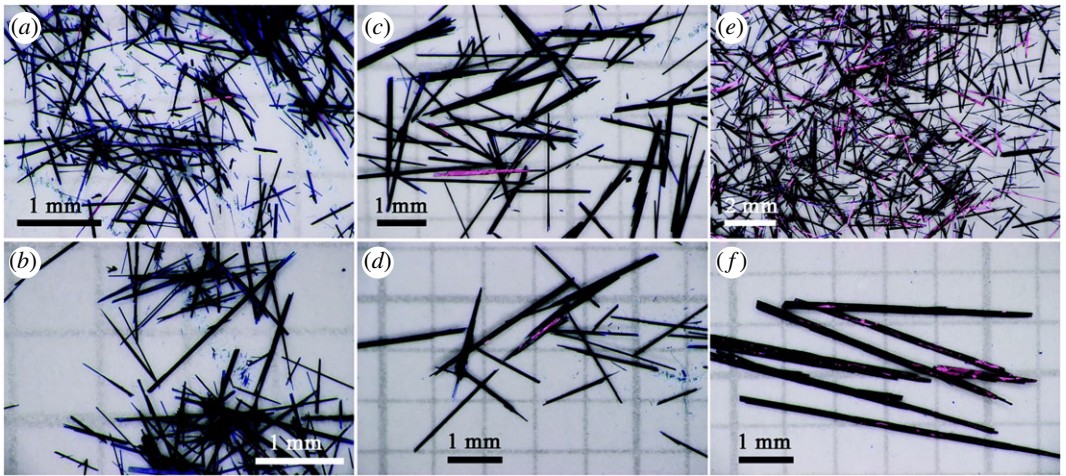

**Figure 2.** The optical images of ZnPc single crystals synthesized from different reaction media: (*a,b*) ethanol; (*c,d*) pentanol and (*e,f*) benzyl alcohol. Adapted from Royal Society of Chemistry.

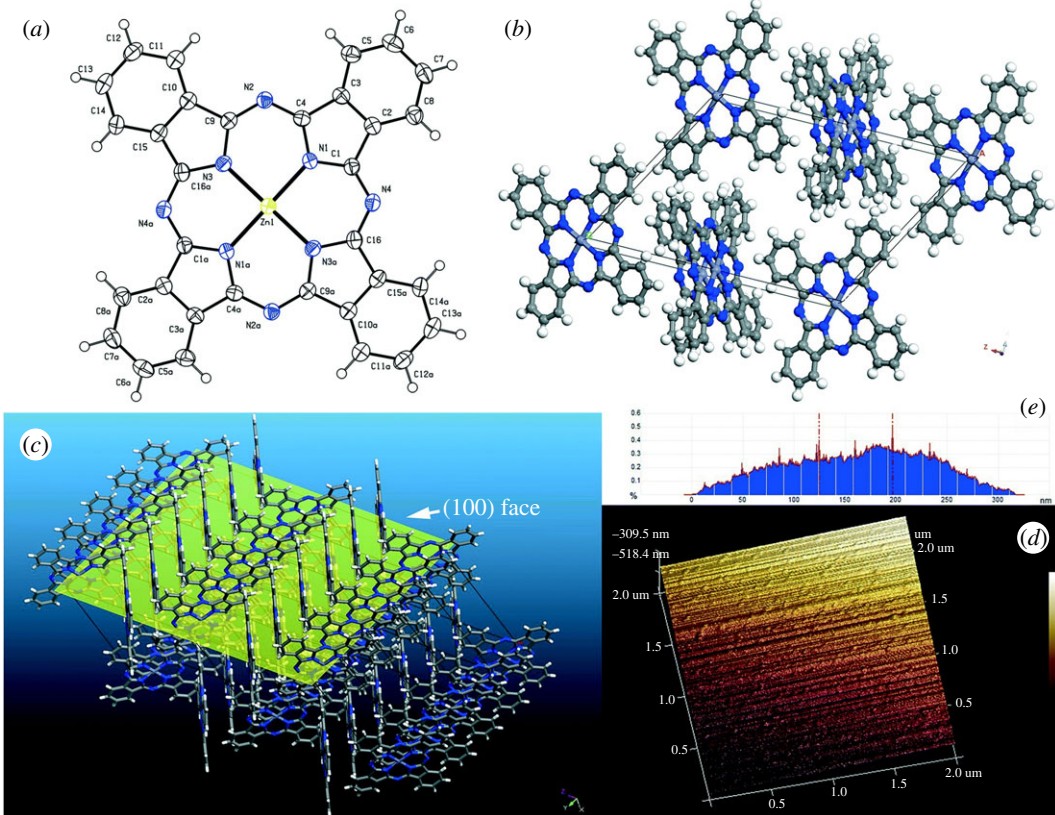

**Figure 3.** (*a*) The crystallographically determined structure of the ZnPc molecule. (*b*) Molecular packing structures in the single crystal of ZnPc according to single-crystal X-ray diffraction. (*c*) Topological view of the predominant (100) face of ZnPc crystals. (*d*) Atomic force microscopy recording on a large and smooth surface of a ZnPc single crystal selected from the benzyl alcohol system. (*e*) Depth histogram based on the AFM morphology analysis of (*d*). Results of depth analysis: peak-to-peak distance, 71.9578 nm; minimum peak depth, 124.182 nm; maximum peak depth 196 nm; depth at histogram maximum, 196.139 nm; number of peaks found, 138. Adapted from Royal Society of Chemistry.

routes for synthesis and purification of crude MOFs. Organic linker molecules from renewable natural products (biomass) are desirable, for example, cellulose or starch (the same for solvents: water or ethanol). All additional unnecessary steps should be avoided, for example, temporary modification of processes or deprotection/protection. Solvents in very small amounts could serve as catalysts in a few cases. Final products should not be stable in the environment (i.e. should be biodegradable instead of

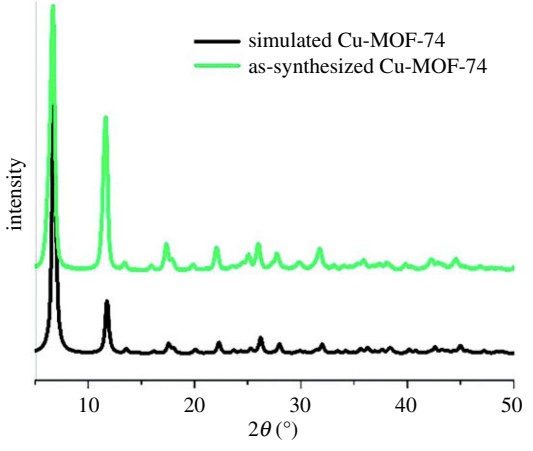
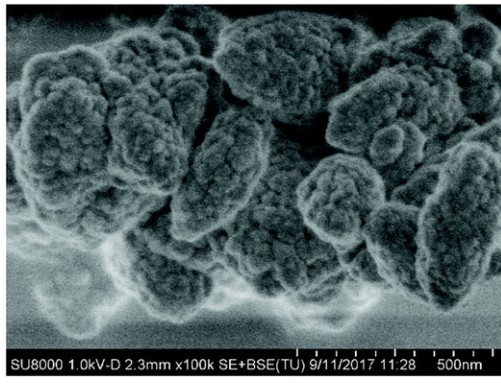

**Figure 4.** Scanning electron micrograph of Cu-MOF-74. Adapted from Royal Society of Chemistry.

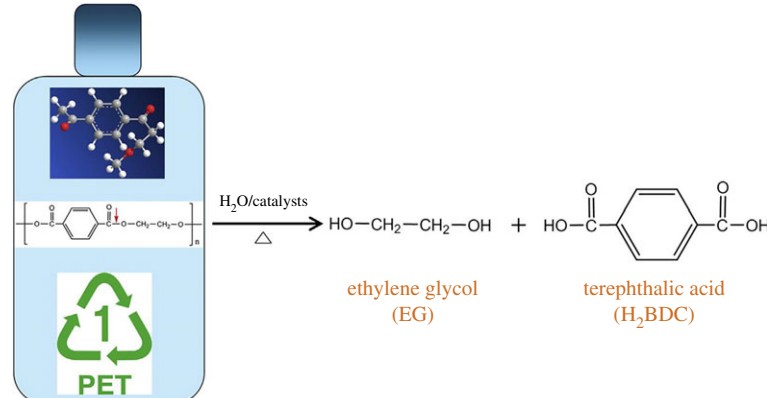

**Figure 5.** The catalysed depolymerization of PET to 1,4-benzenedicaboxylate (BDC) and EG. Adapted from Elsevier.

hazardous or toxic). Processes of MOF synthesis should not be flammable, explosive or have any other type of risk, as well as should be permanently controllable during all synthesis time. Solvo/hydrothermal techniques are preferable. Reaction conditions need to be scalable at least for several kilogram of final products and be carried out preferably at r.t. or maximum in reflux temperature. Classic examples of such greener synthesis of MOFs are the preparation of HKUST-1 (HKUST, Hong Kong University of Science and Technology) or [Cu$_3$(BTC)$_2$(H$_2$O)$_3$] (BTC, 1,3,5-benzenetricarboxylate) via ball milling, ZIF-8 or [Zn(MeIm)$_2$] in methanol, MOF Fe-MIL-101-NH$_2$ or [Fe$_3$O(OH)(H$_2$O)$_2$(O$_2$C-C$_6$H$_3$NH$_2$–CO$_2$)$_3$] from aminoterephthalic acid for 5 min, Zr MOF UiO-66-(CO$_2$H)$_2$ from ZrCl$_4$ in water, among many others.

Nanocrystalline (35–50 nm of crystal size) Cu-MOF-74 (figure 4), containing unsaturated Cu metal sites, can be prepared from 2,5-dihydroxyterephthalic acid (DHTP) and Cu(OAc)$_2$·H$_2$O in methanol at r.t. [93]. It was found that the access to these sites allows using this MOF as a heterogeneous catalyst for conversion of *trans*-ferulic acid to one of the most appreciated flavouring substances, vanillin (97% selectivity, 71% yield). The Cu-MOF-74 possesses low chemical stability in these conditions, being disrupted and transformed to copper oxalate during the catalytic process '*trans*-ferulic acid → vanillin'. This is a natural product, which can be prepared biochemically, but the majority is fabricated chemically, to be used in pharmaceutical and nutraceutical industry, food and cosmetics. Another example of green-synthesized MOF is a high-value Cr-MOF, prepared in one step from waste polyethylene terephthalate (PET) and CrCl$_3$·6H$_2$O in a Berghof high-pressure reactor (figure 5) in water medium at 210°C for 8 h and recommended as a hydrogen-storage material [94].

An Al-MOF 'Al-MIL-68-Mes' (figure 6), containing small trigonal and large hexagonal channels, having high specific surface area 1040 m$^2$ g$^{-1}$, chemical composition [Al(OH)(O$_2$C-C$_3$H$_4$-CO$_2$)]·$n$H$_2$O and derived from mesaconic acid (methylfumaric acid H$_2$Mes), was prepared from mesaconic acid, NaOH and Al$_2$(SO$_4$)$_3$·18H$_2$O in mild conditions at 95°C in water for 45 min under MW [92]. The formed MOF exhibited a considerable chemical and thermal (350°C) stability. Also, enzyme-Zn-ZIF-8

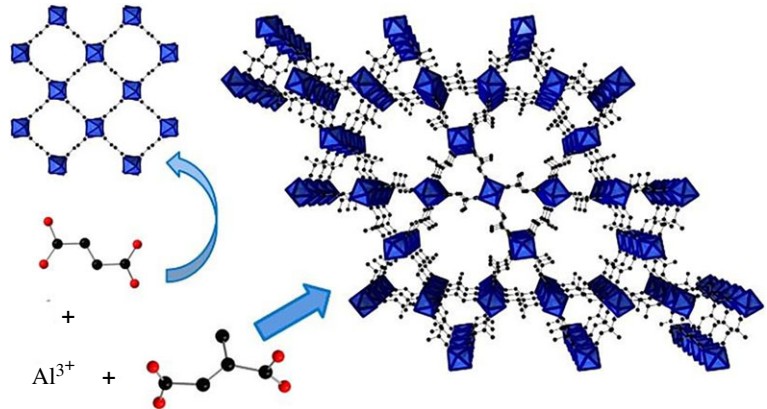

**Figure 6.** Framework structures and linker conformations for Al-MIL-53-Fum and fumarate (top left) and Al-MIL-68-Mes and mesaconate (bottom left and right). Adapted from Wiley.

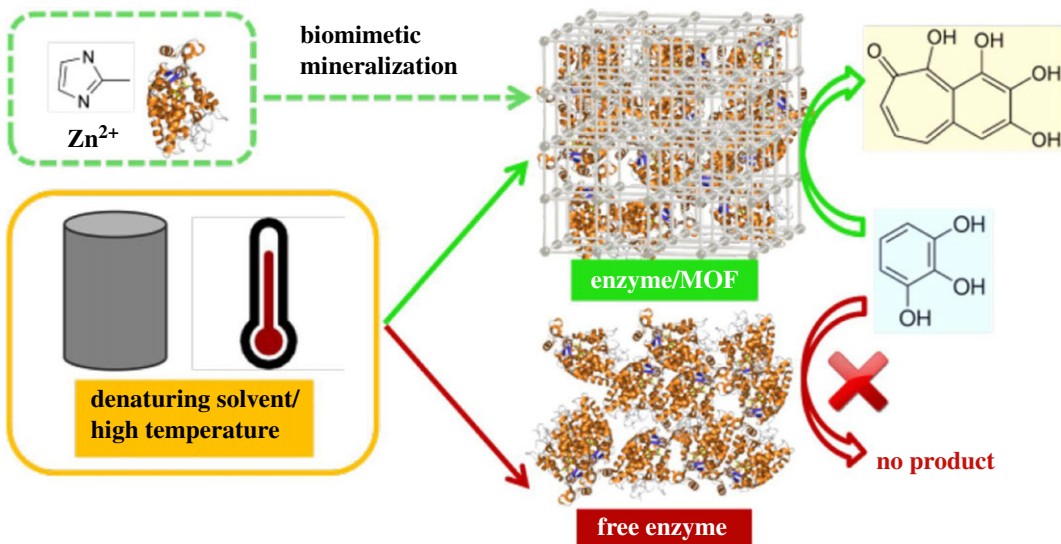

**Figure 7.** Green synthesis of enzyme–MOF composites exhibiting tolerance for denaturing solvents and heat. Adapted from Springer.

MOF composites were prepared (figure 7) by mineralization biomimetic procedure in water at r.t. for 30 min using zinc nitrate and the following model enzymes: horseradish peroxidase (HRP), cytochrome *c* (Cyt *c*), and *Candida antarctica* lipase B (CALB) [95]. ZIF-8 provides a protective layer (shell), growing around the molecule of enzyme after induction of the MOF by protein. After their incorporation into the MOF structure, activities of these protected enzymes remain almost 100% unchanged, in particular when they are in alcohols ($CH_3OH$ and EtOH), DMF or dimethylsulfoxide (DMSO), in a difference with free enzymes, losing 80% of the initial activities.

# 5. Metal nanoparticles

In this section, we will describe the greener methods for obtaining metal nanoparticles. It is worth noting that this field is well developed, first of all, for silver and gold nanoparticles, applying generally biological techniques, lesser for other metals. There are many recent reviews and book chapters in this field; some selected ones are [96–98].

## 5.1. General approach

Preparation of nanoparticles, first of all those of elemental metals, has been the central point of green synthesis processes (together with organic reactions discussed above) for the past 15 years. As general approaches for the nanoparticles synthesis [99], the following aspects are useful in these processes in

order to make them greener. Thus, *capping agents* are used in the majority of syntheses for stabilization, shape control and prevention of aggregation of formed nanoparticles. Classic capping agents in the nanochemistry are long-chain hydrocarbons, functionalized with a heteroatom (oleyamine, trioctylphosphine, oleic acid, dodecanthiol), polymers (polyvinyl alcohol, PEG) and block copolymers (poly(acrylic acid)-block-polystyrene), dendrimers, etc. For green chemistry purposes, the following agents are considered as greener in the synthesis procedures for nanoparticles formation: (i) polysaccharides (like starch or dextran) exhibiting mild capping ability and water solubility (sometimes reduction properties), avoiding toxic solvents and allowing easy separation of nanoparticles from reaction media; (ii) biomolecules (proteins and peptides) possessing high biocompatibility; and (iii) small molecules (i.e. CO) together with organic capping agents. The natural products, used in the synthesis of nanoparticles and nanomaterials, can be applied not only as capping agents, surfactants, solvents and reactants, but also as carriers, catalysts and templates. The use of *ligands* can passivate and coat nanoparticle surfaces, thus stabilizing them, preventing agglomeration and influencing their chemical properties. Typical ligands in the nanoparticle synthesis are phosphines, thiols, amines, selenols, carbenes and alkynyls, being classified by their molecular structures and 'head group'.

Classic *reducing agents* in nanoparticle synthesis are normally $N_2H_4$, HCOH and $NaBH_4$. Ascorbic acid as an organic reductant can be also used. The following greener reductants are recommended for safer obtaining nanoparticles: (i) molecular $H_2$ (disadvantage: combustibility); and (ii) polysaccharides (used also as capping agents, see above) like b-D-glucose, starch or amylose, possessing water solubility and avoiding hazardous organic solvents. $NaAlH_4$ and other strong reductants contribute to the formation of small nanoparticles, whereas plant extracts result in polydisperse products and larger particles. Their use is directly or indirectly related with toxicity, since non-toxic (green) reductants are not so strong in order to be able to form metal nanoparticles of sufficiently high quality, where a much faster kinetics is needed. On the other hand, strong reductants are frequently toxic and expensive. So, one of the main purposes of green synthesis of high-quality nanoparticles is the search for or creation of green (non-toxic) and simultaneously strong reductants.

In the case of *solvents*, whose consumption in some manufactures, for example, pharmaceutical industry, is over 80%, their correct selection is highly important, taking into account the high amount of used solvent wastes and their toxicity. Solvents are applied for dissolution of raw materials/ reactants, heat transfer, dispersion and solubilization of formed nanoparticles. So, the following solvents are recommended as greener media for nanoparticle low- and large-scale synthesis: (i) obviously water as preferred solvent (if, for any reasons, it is impossible to carry out a solvent-less synthesis as the best option), always available at low cost and non-flammable (disadvantage: high heat capacity, inhibiting energy-saving production), and (ii) SC fluids ($CO_2$, $H_2O$ and ILs). In the case of SC water (critical pressure 22.1 MPa, critical temperature 646 K), the SC hydrothermal synthesis is controllable and economically preferable; in addition, SC water is able to dissolve organic substances and to obtain nanoparticles, highly dispersible in organic media. In the case of SC $CO_2$ (SC pressure 74 bar, SC temperature 304 K), it is also promising, being non-flammable, non-toxic, compatible with the environment, chemically inert and inexpensive. ILs, consisting of charged organic and inorganic ion pairs, can substitute toxic and volatile organic solvents; in this case, capping agents are generally unnecessary. The use of ILs can be united with microwave (due to high dielectric constants, high polarities and high ionic charges) and ultrasonic treatment.

Among greener *synthesis strategies* for nanoparticle synthesis, in addition to correct selection of solvents, capping agents and reductants, the selection of heating method is also important to avoid high energy consumption. Furnace, oil/water bath and heating mantle are classic heating sources, which can be replaced by microwaves, whose aid has already led to preparation of a series of nanoparticles (metals, metal chalcogenides, phosphates, oxides, etc.). Solvent-less MW-assisted route is preferable or, in the case of impossibility, the use of DMF or DMSO (sometimes, additional small IL amounts are useful), possessing high dielectric constants, is recommended. Larger nanoparticles are normally synthesized at prolonged MW-irradiation time. Ultrasound-assisted synthesis of nanoparticles is also considered as green method, where the heating is produced from acoustic cavitation and not directly from ultrasound itself. Ultrasonic treatment allows also maintaining formed nanoparticles to remain small without fast further agglomeration. After preparation, nanoparticles are normally separated from reaction medium by *precipitation* and washed by their redissolving and precipitation. Post-processing can include size sorting.

Greener syntheses of nanoparticles, in particular, are carried out by *biological methods* (see also §2.2), where reducing agents are organic compounds and biomolecules (enzymes, formed as a result of cell activities in microorganisms (inside cells or on their surfaces), reduce metal ions to nanoparticles):

(i) microbial synthesis (applying fungi, bacteria, viruses, yeasts and actinomycetes); and (ii) phytosynthesis using plant extracts (now considered as promising for scaled-up fabrication of nanoparticles), avoiding microbial culturing and isolation problems. Mechanism of metal ion reduction using microorganisms is described in [100,101]. Metal nanoparticles can be obtained with the aid of plant extracts, since several phytochemicals are present in leaf extracts: ketones, amides, aldehydes, polyphenols, carboxylic acids, flavonoids and terpenoids [102,103]. Metal nanoparticles, biologically produced using plant extracts, need to be *extracted and purified* after process completion by their transfer to aqueous solution from biomass [104]. Several methods are applied for this purpose, in particular the ultrasound-assisted use of citrate (more effective) or cetyltrimethylammonium bromide (CTAB). It was shown that, in particular, larger Au nanoparticles are extracted after smaller ones. Other methods include enzymatic lysis of the plant cells (expensive and non-scalable method), centrifugation (it could cause aggregation), osmotic shock, heating and freeze–thawing processes, which could be accompanied by sedimentation/precipitation and aggregation, change of size and/or shape. The following disadvantages are typical for the microorganism-assisted synthesis of nanoparticles: (i) long-term maintenance and special facilities for safe microbial growth; (ii) problems in the control of crystallization, shape and size of nanoparticles; (iii) growth of microbes and other biological processes can be affected by temperature and pH changes; and (iv) certain risks (biosafety issues) working with bacteria [105].

Plants, in order to fight against toxic metals and avoid an excess of ROS leading to cell damage, are able to detoxify metals, forming their complexes with flavonoids, phytochelatins, oligopeptides and some proteins by chelation. These antioxidant mechanisms are assisted by phenolic compounds (widespread in many plants), which 'turn on' under increasing metal ion concentration, leading to metal ion reduction upon the contact with organic matter in plant extracts. Resulting nanoparticles possess distinct shapes and sizes, depending on several factors (capping agent, temperature, concentration of metal ion and reductant, pH, etc.). In more detail, these mechanisms are described in a review [106].

Generalization of the important synthesis methods of obtaining metal nanoparticles is given in several reviews [104,107]. In the case of Au, Ag, Pd and Cu, all of them have been obtained by chemical reduction and biological methods. Additionally, the following techniques were applied alone: Au (UV light, laser ablation, lithography, ultrasonic treatment, aerosol methods), Ag (thermal, photochemical, microemulsion, and electrochemical methods, laser ablation), Pd (thermal methods, sonochemical reduction, reactions in SC $CO_2$), Cu (thermal and microemulsion methods, radiation-assisted techniques, mechanochemical procedures and laser ablation).

## 5.2. Noble metal nanoparticles

Nanoparticles of metallic silver and gold have been prepared by green methods much more in comparison with other noble and other metals with aid of bacteria, fungi, plant extracts, yeasts, viruses and algae. In the synthesis of Ag nanoparticles, a variety of the following *physical methods* [108,109] (accompanied by chemical reduction of $Ag^+$ ion to convert it to $Ag^0$) have been applied: UV- and sunlight photoreduction (leading mainly to large nanoparticles and nanocrystals; long-term acting stabilizing agents required), ultrasound-assisted processes (temperature control is needed), laser ablation (leading to nanoparticles with perfect shapes and sizes), microwave-assisted reactions (stabilizing agents required), radiolysis, electrochemical reduction and high-pressure (autoclave) technique. Physical methods, used for preparation of silver nanoparticles, have the following disadvantages [108,110]: tube furnace (large space, heating environment), ceramic heater (fast cooling needed), laser ablation (depends of laser characteristics) and arc discharge (use of Ag wires as electrodes). Also, $Ag^0$ nanoparticles were prepared by chemical routes [108] using the following reductants, among others: $NaBH_4$, sodium citrate, ascorbic acid, curcumin, heparin, polyblock polymers and polysaccharides, as well as thermal decomposition of $AgNO_3$ via spray pyrolysis. These reactions are different in respect of time (from minutes to 7 h), pressure and temperature and lead to distinct sized of $Ag^0$ nanoparticles.

As example of green synthesis of noble metal nanoparticles, a simple use of ascorbic acid as a reducing agent and sodium carboxymethylcellulose as a structure-directing agent, one-dimensional Ag nanobelts (lengths of 50 μm, figure 8) and other nanostructures (brick-like, pie-like and three-dimensional hierarchical ones, figure 9) were prepared from $AgNO_3$ at large scale in mild conditions at 30°C and used for catalytic purposes [111]. As a result of classic action of polyphenol-type compounds, Au nanoparticles (16.6 nm, diverse shapes) were prepared with the aid of catechin (a polyphenol compound, belonging to flavonoids) as a reducing and capping agent (figure 10) [112].

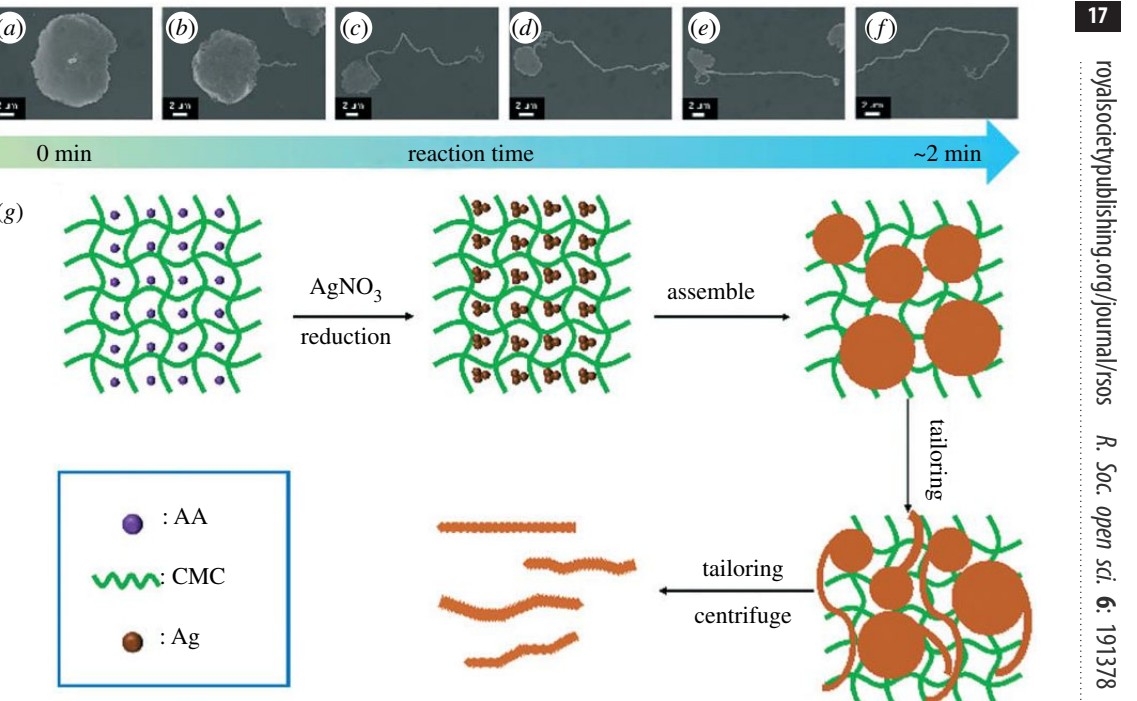

**Figure 8.** Time-dependent intermediate products of the AgNBs (*a*–*f*) and schematic illustration of the growth process of the AgNBs (*g*). Adapted from Royal Society of Chemistry.

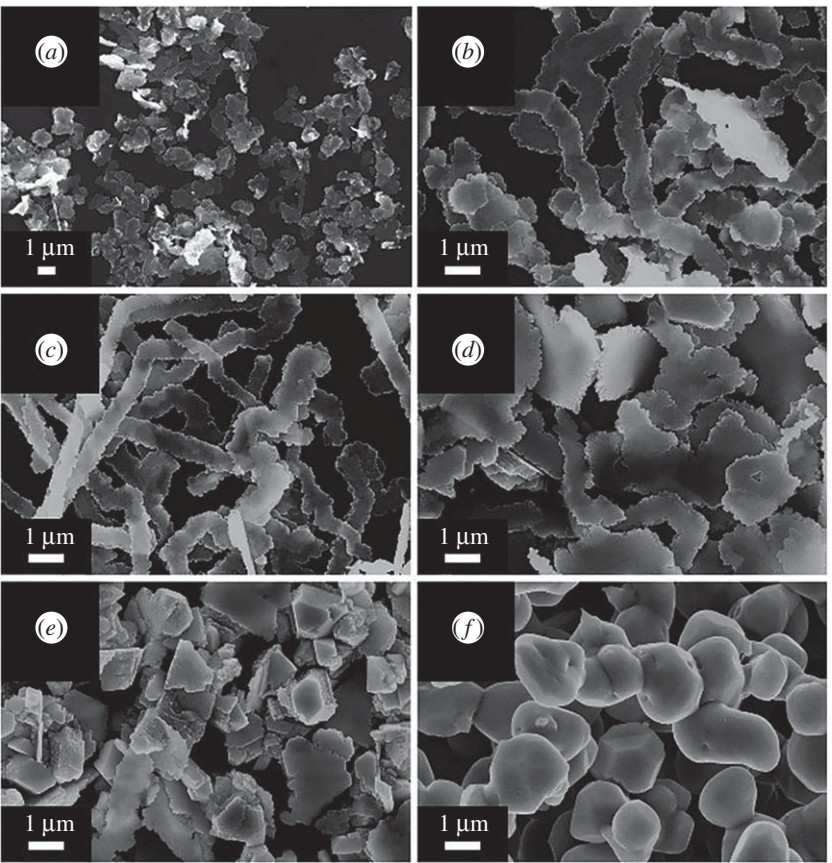

**Figure 9.** SEM images of the silver microstructures obtained with different carboxymethylcellulose concentrations: (*a*) 0.50%, (*b*) 0.30%, (*c*) 0.20%, (*d*) 0.05%, (*e*) 0.01% and (*f*) 0%. Scale bar: 1 μm. Adapted from Royal Society of Chemistry.

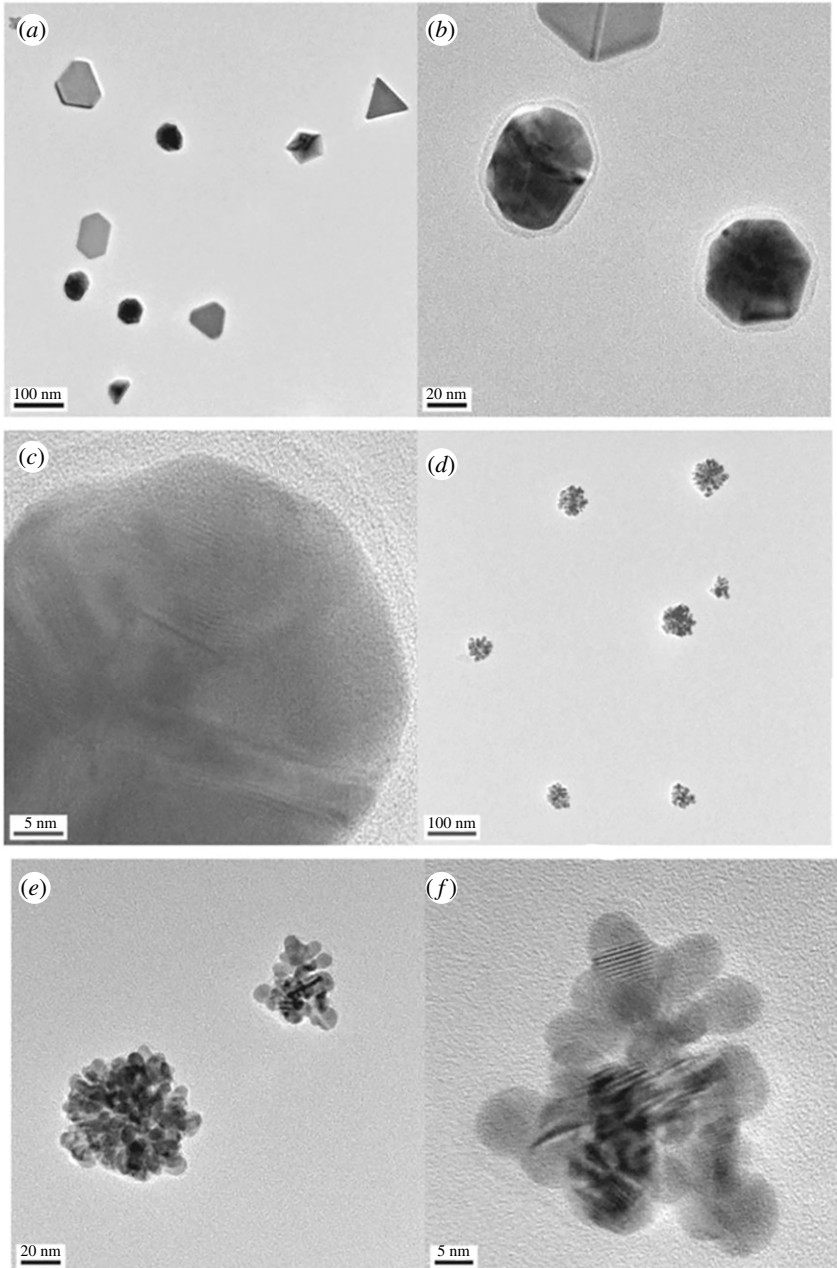

**Figure 10.** High-resolution transmission electron microscopy images of freshly green-synthesized AuNPs. The scale bar represents (a) 100 nm, (b) 20 nm, (c) 5 nm, (d) 100 nm, (e) 20 nm and (f) 5 nm. Adapted from Springer.

In the case of *algae* as green medium [113], both microalgae (*Chlorella vulgaris, Chaetoceros calcitrans, Spirulina platensis, Oscillatoria willel, Plectonema boryanum*) and macroalgae (*Sargassum cinereum, Padina pavonica, Caulerpa racemosa, Ulva lactuca*) were used for $Ag^0$ nanoparticles synthesis. *Plant extracts* have been extensively used for fabrication of Au and especially Ag nanoparticles (interest caused by the antibacterial properties of Ag nanoparticles). In some reactions, *Phoenix dactylifera* seeds [114] (as waste product), *Taraxacum officinale* [115], *Cassia tora* L. roots [116], *C. longa* tuber, soya bean leaf (*Glycine max*) [117] and *Hippophae rhamnoides* (Pd nanoparticles) extracts were applied, among many others. In addition, Ag nanoparticles (from $AgNO_3$) and $TiO_2$ nanoparticles (from $TiO(OH)_2$) were prepared using aqueous leaf extract of *Euphorbia prostrata* [118]. Also, spherical Ag (12 nm) and hexagonal and triangular Au (11 nm) nanoparticles were obtained with the aid of leaf extract of *Rosa rugosa* in water for 10 min [119]. Metal ion and leaf extract concentrations influence considerably the properties of the formed products in this interaction; these factors are critical also in other metal reduction processes using plant extracts.

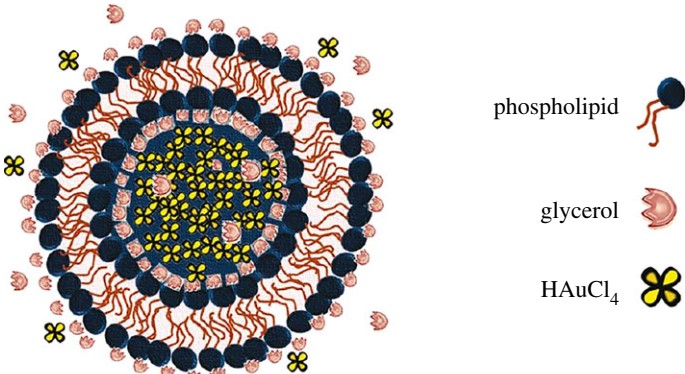

**Figure 11.** Schematic of the designed liposomal nanoreactor. Adapted from ACS Publications.

In addition to plant extracts above, other natural products, as, for example, black and green tea, coffee or honey have been involved in the noble metal nanoparticles formation. Thus, *honey*-assisted synthesis led to metal (Ag, Au, Pd, Pt) and carbon (highly fluorescent carbon dots) nanoparticles, where honey, oldest food source, takes part as reducing and stabilizing agent [120]. This method is faster in comparison with microorganism-assisted techniques. As examples of *fungi*-assisted synthesis, we note Au (6–40 nm) nanoparticles, prepared with the aid of thermophilic fungi [121], and Ag nanoparticles, hydrothermally synthesized using the fungus *A. fumigatus* mycelia extract [122]. It was concluded that fungi are much preferable in comparison with bacteria for the synthesis of metal nanoparticles.

Au nanoparticles (2–8 nm in size) were prepared from $HAuCl_4$ using nanoreactors (figure 11) based on *liposomes* and glycerol as green solvent and catalyst [123]. Being compared with classic methods in microwave and reflux conditions requiring heating, no harsh chemicals were used in this greener procedure, where glycerol is incorporated in both internal and external surfaces of phospholipid layers. The reducing agent remains relatively mobile in liposomal membrane, allowing a controllable nanoparticle growth on the nucleation sites. Au nanoparticle size depends on glycerol concentration, being decreased at its higher content, and temperature, studied for the range 4–50°C. Nanoparticles with other shapes and properties were obtained in the similar conditions without such nanoreactors. It is also possible to obtain monodispersed Au nanoparticles using papaya juice as a capping and reducing agent, with approximate sizes of $6.9 \pm 1.1$ nm at a temperature of 120°C for 3 min [124].

Other metal nanoparticles have been obtained by greener methods (in particular, biologically) in a considerably lesser number of examples. Thus, the following plant extracts have been applied for metal nanoparticle synthesis: *Ocimum sanctum* (Ni) [125], *Camellia sinensis* [126] and *Spinacia oleracea* [127] (Fe), *Solanum lycopersicum* (tomato aqueous extract) for obtaining dispersed and highly stable (for 80 days) Cu nanoparticles (40–70 nm) [128]. In the case of copper, spherical Cu nanoparticles (20–30 nm) were also obtained from $CuSO_4$ as copper source using acidic chitosan solution at 70°C [129]. These nanoparticles were found to be able to combat pathogenic microorganisms.

Manganese nanoparticles (50 nm) were prepared using lemon extract as reducing agent. As a stabilizing agent, the turmeric curcumin was used. The obtained Mn nanoparticles possess antimicrobial and inhibition activities against several microorganisms, including *E. coli*, *Staphyloccus aureus* and *Aspergillus niger* [130]. In the case of lead, the colloidal Pb nanoparticles (47 nm) can be prepared from lead acetate using agricultural waste *Cocos nucifera* L. extract (figure 12) [131]. Finally, core–shell and bimetallic nanoparticles possessing technical (removal of heavy metals) and biomedical applications (microbial and molecular sensors, in parasitology, for oligonucleotide and bacterial detection, as gene carriers, etc.) are recently reviewed [132].

## 6. Non-metal elemental nanoparticles

This section is devoted to the elemental nanoparticles of non-metals, synthesized by greener routes. Practically, all reports on non-metals in elemental state, obtained by green chemistry methods, are related with carbon nanosize allotropes: nanodots with fluorescent properties, nanotubes, graphene and its oxidized forms, as well as some hybrids. Selected recent reviews in this field are described in [133,134]. These nanoparticles have been mainly prepared by hydrothermal, microwave, pyrolysis of plants or their wastes, plant-extract-assisted techniques or their combinations. Varieties of these

**Figure 12.** Possible mechanism of the formation of lead nanoparticles.

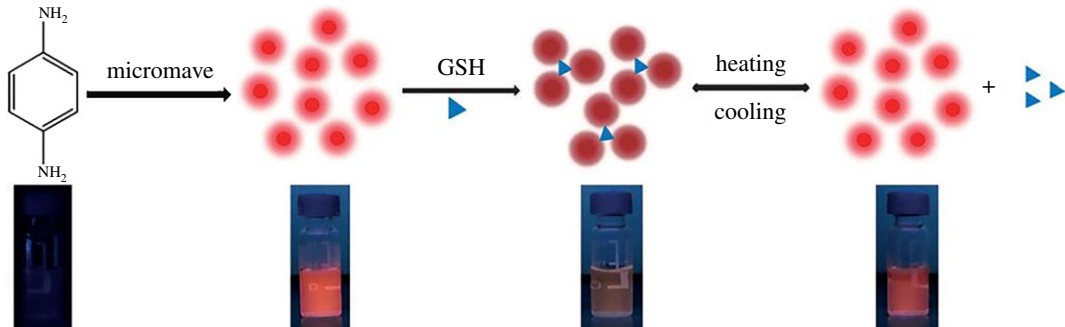

**Figure 13.** Schematic diagram for preparing red-emitting CNDs for the fluorescence 'on–off–on' sensing of GSH and temperature. Adapted from Wiley Online Library.

combinations are especially well seen in the example of carbon dots/QDs, additionally providing their water-solubility properties. Thus, fluorescent carbon nanodots (CNDs, red emission peak at 615 nm, quantum yield 15%) were synthesized by microwave treatment of a non-fluorescent $p$-phenylenediamine in EtOH-$H_2O$ solution leading to its carbonization and formation of CNDs (figure 13) [135]. These CNDs serve as chemosensors for glutathione (GSH) detection and temperature sensor at a molecular level. Alternatively, MW-assisted pyrolysis of reagents was applied for preparation of highly fluorescent N-doped carbon dots of 5 nm size from sesame seeds (8.02% quantum yield) [136], exhibiting water-soluble properties and suitable for Fe(III) selective sensing.

Aqueous fluorescent carbon dots (size range 3.14–4.32 nm) were hydrothermally synthesized from turmeric and black pepper, red chili, cinnamon and sweet potato peels [137]. As a result, it was confirmed that the C dots' toxicity strongly depends on the method of their fabrication, being studied in human cancer cells (gliobastoma) in a range (0.1–2 mg ml$^{-1}$) of concentrations. Red chili-derived C dots showed higher toxicity, being compared with citric acid-derived C dots, revealing that the selective citoxicity is caused by the presence of functional groups on their surface. The obtained self-fluorescent C dots can be tracked inside cells and exploited for imaging *in vitro*. Water-soluble fluorescent carbon quantum dots (CQDs) (quantum yield 46.6%) were hydrothermally prepared (figure 14) also from *Tamarindus indica* leaves [138] and applied for $Hg^{2+}$ sensing in the range from 0 to 0.1 mM. These bio-compatible CQDs can be potentially applied in diagnostics of diseases, bio-imaging, sensing, etc. As examples of using other plant extracts, fluorescent water-soluble N/P co-doped CQDs were hydrothermally prepared at 90–150°C from edible *Eleocharis dulcis* [139], resulting in easily scalable CQDs serving as sensors for $Fe^{3+}$ ions and having potential applications in

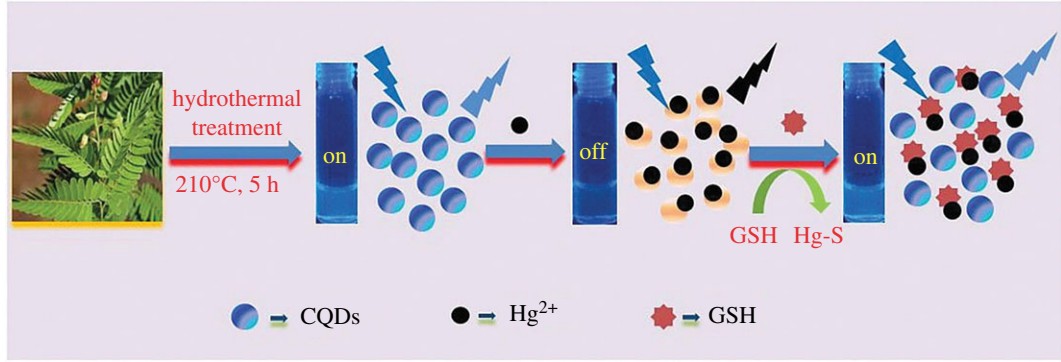

**Figure 14.** Schematic of the synthetic process for the CQDs via a simple one-step hydrothermal treatment using *T. indica* leaves and their application in the turn-off and turn-on sensing of $Hg^{2+}$ and GSH. Adapted from Royal Society of Chemistry.

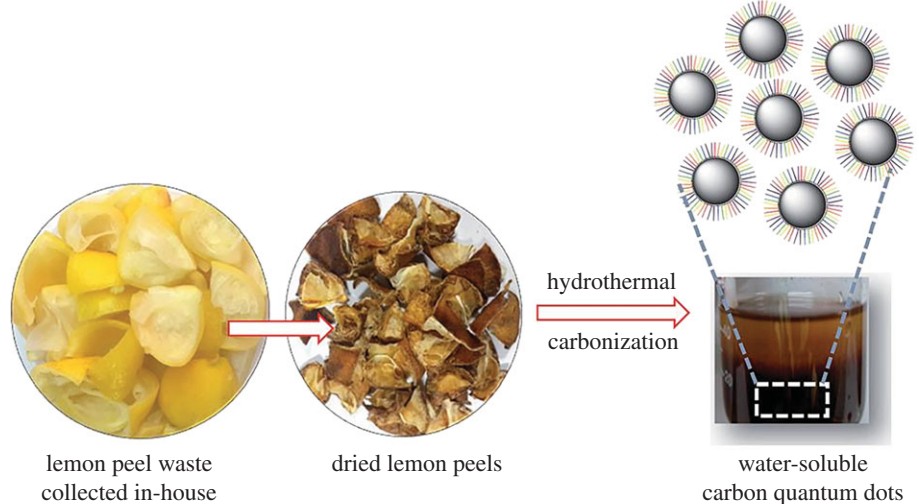

**Figure 15.** Schematic illustration for the synthetic procedure of water-soluble CQDs by hydrothermal treatment of lemon peel waste precursor. Adapted from Royal Society of Chemistry.

an anti-counterfeit area. The *Opuntia ficus-indica* extract, having hydrophilic properties, was used for ultrasound-assisted preparation of laminar carbon from commercial graphite as a precursor for 30 min [140]. Small QDs were obtained from the same precursor by stirring for 30 min at 50°C and further ultrasonic treatment for 30 min more. In addition, water-soluble photoluminescent CQDs (size 1–3 nm, quantum yield 14%) were hydrothermally obtained (figure 15) from lemon peel waste [140] and applied for $Cr^{6+}$ detection (limit 73 nM). This fabrication method can be scaled up. The composite of these CQDs with $TiO_2$ is capable to carry out photocatalytic degradation of MB dye, having catalytic activity 2.5 times more being compared with $TiO_2$ nanofibres due to a better charge separation on the interface in the hybrid, and the sweet potatoes are considered as waste materials from which carbon nanodots with a diameter of $2.0 \pm 0.6$ nm can be obtained with a quantum yield of 8.9%, which when coated with (3-aminopropyl) triethoxysilane to be used as an ecofriendly sensor for oxytetracycline detector, with a detection limit of 15.3 ng ml$^{-1}$ [141].

In the case of graphite oxide (GrO)-graphene(G)-GO conversions, active substances in plant extracts lead to the reduction of GO or GrO forming rGO or graphene. Thus, graphene was prepared by GrO reduction using *Cocos nucifera* L. (coconut water) under sonication for 1 h and further heating at 80–100°C for 12–36 h [142]. rGO nanosheets, functionalized with polyphenol, were prepared according to the Hummers method by action of *Citrullus colocynthis* leaf extract as stabilizing and deoxygenating agent [143].

In these conditions, rGO nanosheets are not subjected to further agglomeration and, in addition, were found to possess a dose-dependent toxicity against DU145 cells. Extracts of carotenoids available in vegetables (sweet potato, carrot, etc.) were used for GO reduction in aqueous NaOH medium at 90°C under sonication, leading to the $sp^2$ carbon atom restoration [144]. Also, structural changes of GO shape can occur. Thus, GO tubes of micrometre length, having irregular three-dimensional network

forms, were obtained by a metal-complexation route by GO cross-linking with Zn, Ca and Sr chlorides in water in a temperature range 24–55°C for 2 h [145]. $Ca^{2+}$ coordination required a higher temperature than $Zn^{2+}$ and $Sr^{2+}$. Such composites can be applied in the biomedical field, for example, GO tubes and sheets with calcium alginate, showing a considerable mechanical improvement in comparison with GO.

There are several reports on the growth of carbon nanotubes (CNTs) using plant extracts as catalysts; however, high temperatures are requiring for their formation. Thus, CNTs were grown using rose (*Rosa*), garden grass (*Cynodon dactylon*), walnut (*Juglans regia*) and neem (*Azadirachta indica*) plant extracts in methanol, loaded in a Si sample, as catalysts by chemical vapour deposition (CVD) of acetylene gas at 575°C; meanwhile, carbon nanobelts and SWCNTs were obtained at 800°C (not in cases of all extracts) [146]. The products did not contain toxic metals; this is a great advantage of this greener method. As an example of green-chemistry-produced CNT composites, highly thermally stable carbon nanotubes/ polyaniline (CNT/PANI) hybrids, useful for modification of electrodes on nickel foam basis, were prepared by aniline polymerization of aniline in multi-walled carbon nanotube (MWCNT)-COOH presence in water and ionic liquid (1-butyl-3-methylimidazolium tetrafluoroborate, [bmim][$BF_4$]) as green solvents and mineral acids (HCl and $HNO_3$) [147]. Polymer layer was found to be coated with CNTs.

Other non-metal nanoparticles are rare, for instance, 28 nm P nanoparticles obtained from tricalcium phosphate ($Ca_3P_2O_8$) as precursor salt by employing *Aspergillius tubingensis* TFR-5 [148], Si and Se nanoparticles, which (as well as Au (from $HAuCl_4$) and Ag (from $AgNO_3$)) [149] were prepared in aqueous media from $Na_2SiO_3$, and $Na_2SeO_3$ as precursors using intra- and extracellular extracts of *Lentinus edodes*, xylotrophic basidiomycetes *Pleurotus ostreatus*, *Grifola frondosa* and *Ganoderma lucidum* [150]. The formed nanoparticles possess size, shape and aggregation grade depending on the extract type and fungi. It was found that, in the case of non-metals (Si and Se), the process does not depend on the activity of phenol oxidase, in a difference with metals (Ag and Au). In the case of $Na_2SiO_3$ use, silicon nanoparticles were observed using extracellular extracts, being large or small, depending on the extract. These processes can be controlled by variation of conditions, leading to necessary parameters of the final nanoparticles.

# 7. Metal and non-metal oxide nanoparticles

In this section, we will discuss oxides of metals and non-metals in the nanostructurized form, obtained by greener methods. Some recent reviews on greener synthesis of ZnO nanoparticles are described in [151–153]. The number of reported nanoparticles, obtained by green chemistry methods, is considerably less, being compared with noble metal nanoparticles discussed above, and practically all of them are metal oxides. Some important methods of their green fabrication (additionally to the microemulsion and biological ones) are the following for synthesis of: zinc oxide (vapour phase oxidation, sol–gel, sonochemical, hydrothermal, precipitation, polyol methods, thermal vapour transport and condensation), magnetite (thermal decomposition, coprecipitation, hydrothermal synthesis, continuous-flow technique) and indium oxide nanoparticles (sol–gel, thermal decomposition, spray pyrolysis, tribochemical technique, hybrid induction and laser heating, pulsed laser deposition, hydrothermal method) [104]. In several processes, including the use of plant extracts, the hydroxides are formed first, then are decomposed by calcination/annealing.

CaO nanoparticles were obtained using papaya leaf extract and green tea extract from calcium nitrate and further calcination of the product ($Ca(OH)_2$) at 500°C for 3 h [154]. Resulting papaya-derived CaO nanoparticles exhibited higher photocatalytic activity for dye degradation, being compared with tea-derived analogue; both products are sensitive to both Gram-positive and Gram-negative bacteria. MgO nanoparticles were synthesized from betel leaf extract without any other reducing and stabilizing agent and further annealing at 400°C [155]. In the case of ZnO nanoparticles, their green chemistry synthesis methods are more variable, taking into account high attention to this compound, its importance, properties (photochemical, catalytic, antimicrobial and antifungal) and applications. The following plant extracts have been applied for obtaining ZnO nanoparticles (from 15 to 70 nm in size), among others [151]: *Vitex negundo*, *Trifolium pratense*, *Lagenaria siceraria*, *Passiflora caerulea*, *C. sinensis*, *Eucalyptus globulus*, *A. betulina*, *Al. vera*, *Pelargonium zonale* leaf, among others [153,156]. Additionally, ultrasound-assisted synthesis from $Zn(NO_3)_2$ and NaOH led to semicristalline ZnO nanorods with layered structure [157].

CuO nanoparticles, prepared by green methods, have been used as antimicrobial, antioxidant and anti-cancer agents, photocatalysts, for dye decolorization (for instance, methylene blue). CuO can be used in

textiles, plastics, coatings, etc. as a chemically stable and long-term antibacterial additive [158]. CuO nanoparticles have been prepared using several plant extracts, in particular *Abutilon indicum*, and possess higher activity than, for example, Ampicillin, against *S. aureus* and *E. coli*. Natural polymers can also result in several metal and metal oxide nanoparticles. Thus, CuO microparticles with 180 nm size, stable for six months and showing anti-algae activity, were prepared from $CuCl_2 \cdot 2H_2O$ using the natural polymer *gum karaya* as stabilizing agent [159]. The same method was also used to obtain nanoscale Pd (1.5 nm), Pt (12 nm), Au (42 nm) and Ag (5 nm), where this polymer acted also as a reducing agent. The antibacterial and adsorption properties of CuO nanoparticles can vary depending on their synthesis method, being more effective when synthesized by chemical precipitation, compared to the microwave and hydrothermal method, due to the increase in the surface area of the synthesized CuO [160].

The magnetite nanoparticles ($Fe_2O_3$) were prepared from $Fe^{3+}$ and $Fe^{2+}$ salts (2 : 1 molar ratio) using *Anthemis pseudocotula* extract [161], and, after functionalization, were used for capture of heavy components in crude oil. Doped iron oxide nanoparticles can also be obtained from plant extracts. Thus, superparamagnetic $Fe_3O_4$ nanoparticles were hydrothermally prepared with aid of pepper extract as a capping agent, followed by the use of the same extract for Pd nanoparticle formation as a reducing agent [162]. Resulting Pd-$Fe_3O_4$ nanoparticles were found to have photocatalytic properties. PbO nanoparticles were chemically (more than 200 nm in size) and biologically (78 nm in size) obtained [163], using $Pb(OH)_2$ dehydration at 320°C and the treatment of lead nitrate with *Bacillus toyonensis*, respectively. Also, $ZrOCl_2 \cdot 8H_2O$ was used as a precursor for preparation of $ZrO_2$ nanoparticles in ethanol medium at 50°C using *Az. indica* leaves extract with further calcination at 600°C. For this biologically obtained product, the band gap was found to be 5.80 eV, which is higher, being compared with the bulk band gap (5 eV).

$CeO_2$ nanoparticles, possessing properties as oxidase, catalase and multienzyme, and, apart, able to self-regenerate their surface (regenerating antioxidant activity), widely applied in biomedicine, bioanalysis and drug delivery, can be prepared by several biological methods, reviewed in [164]. Their phytosynthesis has been reported with the use of plant extracts such as *Al. vera*, *Acalypha indica* and *Gloriosa superba*, leading to relatively large-size crystals having high surface area and antibacterial activity. Among other examples of yeast- and fungi-assisted synthesis, the fungus *Curvularia lunata* was used. $CeO_2$ nanoparticles, obtained by some mycosynthesis methods, where extracellular compounds (enzymes) in fungi act as capping agents, are water soluble, capable to generate ROS, diffusing across membranes and cell walls and leading to their damage. In addition, the $CeO_2$ synthesis can be nutrient-mediated, using natural materials such as honey and especially egg white, in which their proteins lead to the formation and growth of stable small $CeO_2$ nanoparticles. Honey-mediated method can be considered as a sol–gel process, where vitamins, enzymes and carbohydrates with $NH_2$- and OH-groups facilitate the chelation of metal ions, not allowing further agglomeration of formed metal oxide nanoparticles. Also, $CeO_2$ can be obtained using biopolymers, such as agarose polymers as stabilizing/capping agent, changing their solubility in water when being heated/cooled and forming nanochannels (pore size of 20 nm), where $CeO_2$ formation takes place. Starch, dextran, PEG and chitosan are other examples of biopolymers, used for $CeO_2$ preparation. All these methods perfectly fit into the field of greener synthesis, since several biopolymers and plant extracts possess properties such as antibacterial activity, biocompatibility, non-toxicity and biodegradability.

Silica nanoparticles (7–80 nm in size) were prepared with aid of the plant *Cynodon doctylon* as a silica source [165]. In comparison with classic methods from tetraethylsilicate or silicon tetrachloride, this technique does not use chemicals and calcination procedure. The plant stores silica in epidermal parts in the form of phytoliths. In a difference with the biosynthesis of other oxides and especially elemental metals, where plants have been used as reducing and capping agents, here we emphasize a completely different role: nanoparticles' source.

# 8. Nanoparticles of other chemical compounds

In this section, we will describe main groups of nanosized inorganic compounds, obtained by greener methods, which are metal salts with O-containing anions, sulfides and selenides of transition metals. Reviews in this field are scarce [166]. One-dimensional nanofibres and two-dimensional atomic-size orthorhombic thin sheets of boehmite (γ-AlOOH) were found to be annealed to cubic alumina (γ-$Al_2O_3$) [167]. In this green synthesis, the solvent for Al metal in the form of liquid metal (Ga) was used, being then exposed to liquid $H_2O$ or its vapour (figure 16), resulting in $H_2$ bubbles responsible for delamination of thin sheets. Based on these high surface boehmite sheets, the membranes for

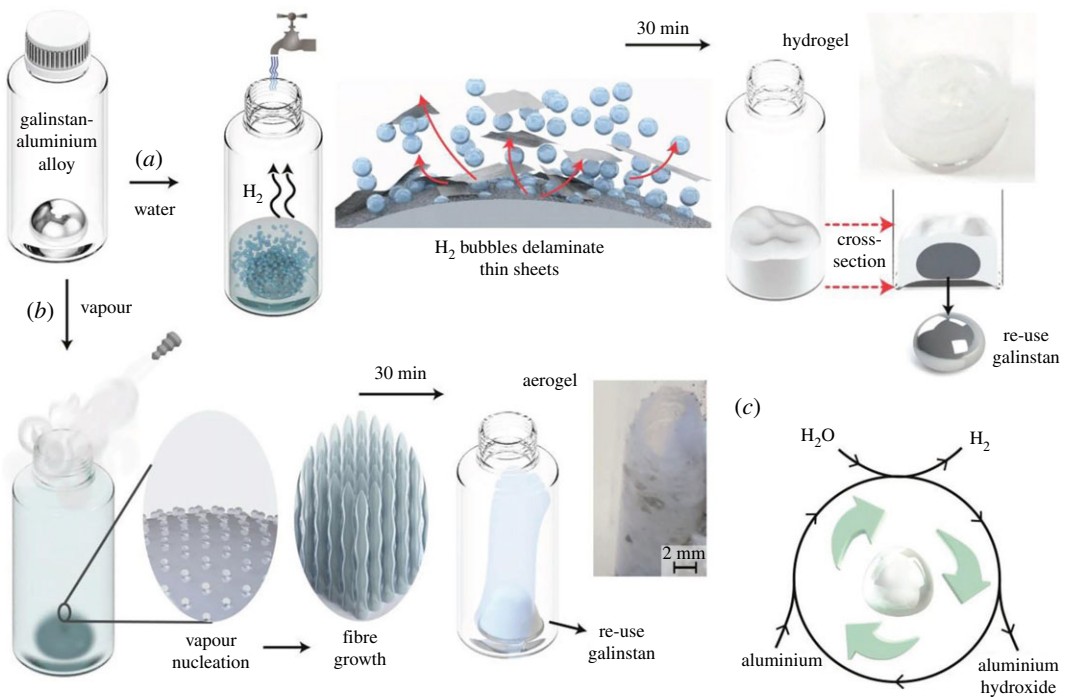

**Figure 16.** (*a*) Deionized water (DI water) is added to a drop of a liquid metal alloy of galinstan-Al. After 30 min of exposure to water, a hydrogel is formed. The proposed growth mechanism of the product shows that the skin is delaminated by hydrogen bubbles in the form of sheets into the DI water. The photo shows the final product from adding 2 ml DI water to 0.5 g of a 3.3 wt% galinstan-Al alloy. (*b*) Vapour is fed into the glass vial containing 0.5 g of 3.3 wt% galinstan-Al. After 30 min, an aerogel has grown from the liquid metal. The growth mechanism shows the emergence of fibrous structures from the vapour nucleation sites. The photo shows the aerogel that has grown from the surface of the liquid metal. (*c*) The reaction stops after the consumption of all the aluminium but the galinstan remains unchanged after producing the aluminium hydroxides. Therefore, a green process is adopted to re-use the galinstan for the next synthesis cycle. Adapted from Wiley.

separation of oil and heavy metal ions from water were offered. LaAlO$_3$ nanoparticles (32–45 nm) of the perovskite-type were synthesized by a molten salts technique from Al(NO$_3$)$_3 \cdot$ 9H$_2$O and La(NO$_3$)$_3 \cdot$ 6H$_2$O nitrates and alkali metal (Li, K) hydroxides at the temperature above melting points of nitrates and firing [168]. This simple method is fast and cost effective, since no expensive chemicals (for instance, alkoxides) are used. Also, dark green NiSnO$_3$ nanopowder with a strong Sn–O–Ni framework and simultaneous bimetallic properties was prepared from SnCl$_2 \cdot$ 2H$_2$O and NiCl$_2 \cdot$ 6H$_2$O in KOH medium by refluxing under stirring at 80°C and further calcination at 1000°C [169]. This material was used for fabrication of a screen-printed electrode for hydroquinone sensing, showing a sensitivity of 6.03 µA mM$^{-1}$. In addition, nanosized CaTiO$_3$, a dielectric ceramic material with perovskite structure, was prepared by a top-down mechanochemical method using a high energy micronizer ball mill in the absence of any solvent from TiO$_2$ and CaCO$_3$ in 1:1 molar ratio for maximum 5 h and further calcination at 800°C for 2 h [170]. This synthesis technique is not so complex as conventional methods (inorganic–organic solution route, hydrothermal, sol–gel methods) and is greener, allowing solvent-less procedure. The milling procedure favours the formation of purer product.

Calcium phosphates, conventionally prepared by sinterization of precursors, hydrothermal or sol–gel methods from CaO, Ca(OH)$_2$ and CaCO$_3$ as calcium source, were obtained by precipitation methods for up to 24 h, varying pH (from 3 to 12), precursors (calcium nitrate, acetate or chloride) and ratios of ion sources (Ca/P = 1–3) [171]. In the case of pH 12, CaCO$_3$/apatite biphasic system was observed. Among other results, it was revealed that alkaline pH contributed to the formation of nanosize spherical shape, meanwhile acidic pH yielded microsize 'columns'. Also, pure barium stannate with cubic perovskite structure (similar to the CaTiO$_3$ above) and average particle size 52 nm was synthesized in two steps from BaCO$_3$ and SnCl$_2 \cdot$ H$_2$O as precursors using the *Al. vera* extract and further calcination at 850 and 1200°C (better purity product). The first step consisted of the preparation of nanosized SnO$_2$ with plant extract by sol–gel interaction, followed by calcination of its mixture with BaCO$_3$. The product can be applied as a ceramic material.

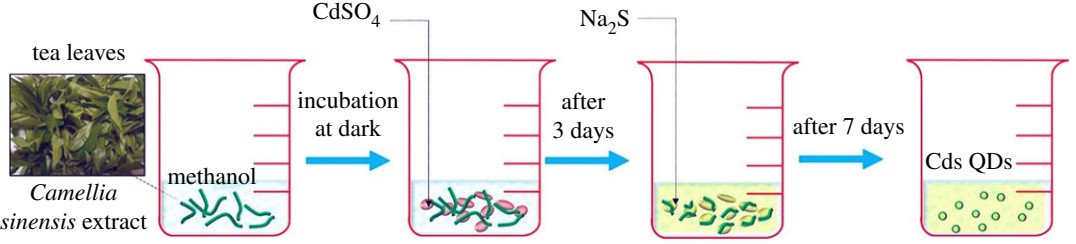

**Figure 17.** Illustration of experimental stages involved in *C. sinensis* extract-mediated green CdS QDs synthesis. Adapted from ACS Publications.

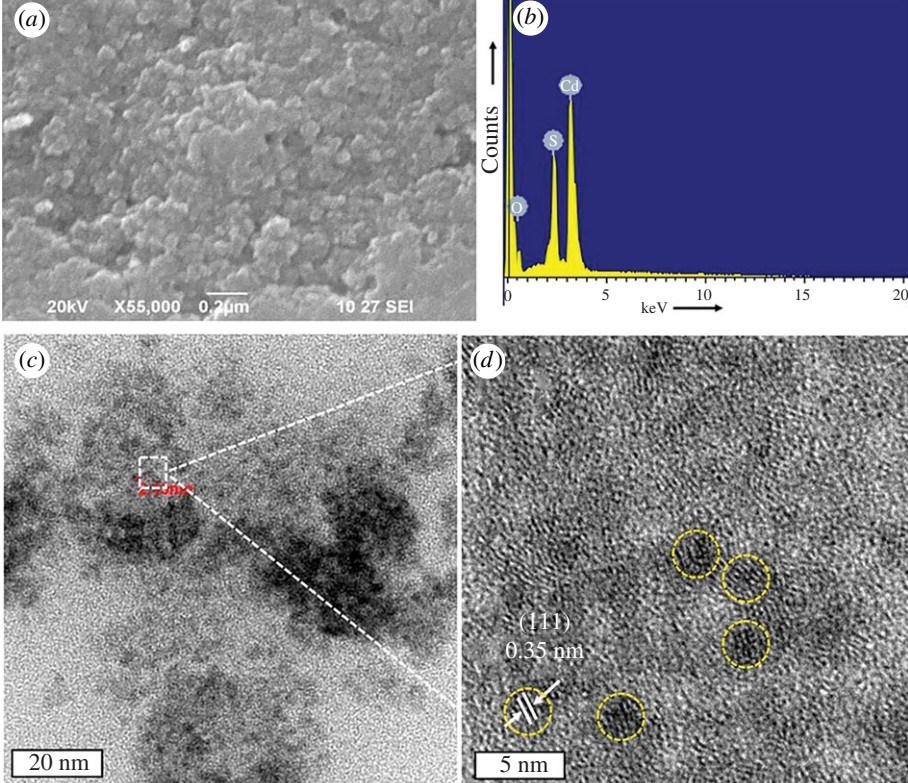

**Figure 18.** (*a*) SEM image of CdS QDs; (*b*) energy-dispersive X-ray spectroscopy (EDAX) spectrum of CdS QDs; high-resolution transmission electron microscopy image of synthesized CdS QDs (*c*) at 20 nm scale and (*d*) high magnification at 5 nm scale. Adapted from ACS Publications.

Another important group of compounds is *metal chalcogenides*. In addition to CQDs described above, S-based *QDs*, colloidal fluorescent semiconducting nanocrystalline materials, which are applied in a variety of areas such as drug delivery, nanomedicine, photovoltaics, biosensing, molecular pathology, *in vivo* imaging techniques, etc. can be fabricated by green methods [166]. Thus, heavy-metal-free InP/ZnS core/shell QDs (4 nm average diameter, quantum yield of 60.1%) with maximum fluorescence peak at approximately 530 nm were solvothermally synthesized from $InI_3$ and $ZnCl_2$ as metal-containing precursors at optimal temperature 70°C [172]. Relatively water-soluble lead sulfide QDs were prepared using GSH and 3-MPA (mercaptopropionic acid) as the stabilizing/capping ligand under 30 min sonication [173]. These capped QDs were then functionalized with streptavidin and further bound to biotin. The formed products were found to be suitable for deep tissue imaging, since they do not affect the normal cells and so are useful for cellular tracking in the human body. Also, ultrasmall semiconductor CdS QDs (2–5 nm in size), useful for biomedical applications, were prepared from $CdSO_4$ as metal source and $Na_2S$ using tea leaf extract (*C. sinensis*) as a capping agent (figure 17) [174]. The obtained fluorescent CdS QDs (figure 18) are stable in the bioenvironment and provide effective intracellular cell tracking in cancer cells (figure 19), being able to penetrate and degrade them. Another interesting aspect is that mother leaves, in addition to fresh leaves, can be used for extract preparation, which makes this technique as cost-effective.

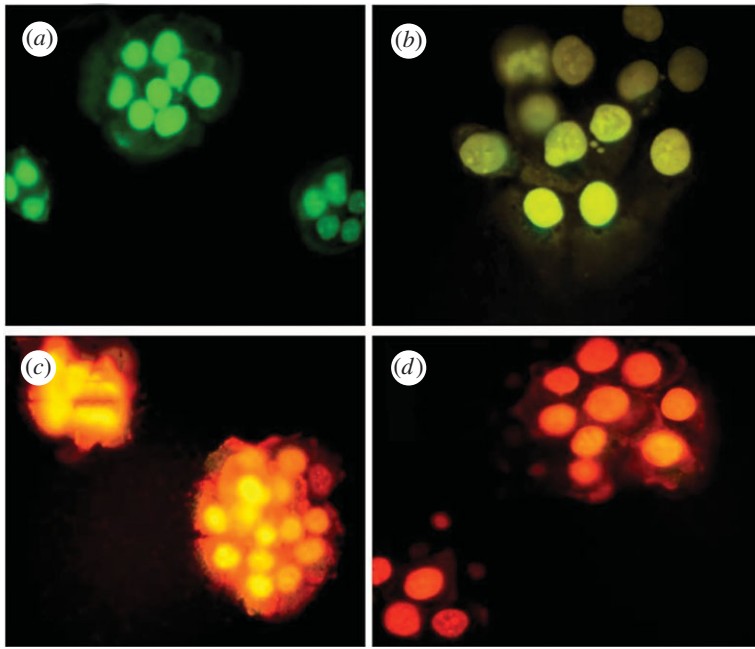

**Figure 19.** Fluorescence microscopy images of AO/EtBr-stained A549 cells (*a*) untreated (control) and; CdS QDs treated at different concentration (*b*) 10, (*c*) 25 and (*d*) 50 µg ml$^{-1}$. AO/EtBr images are recorded using excitation at 460 nm. Adapted from ACS Publications.

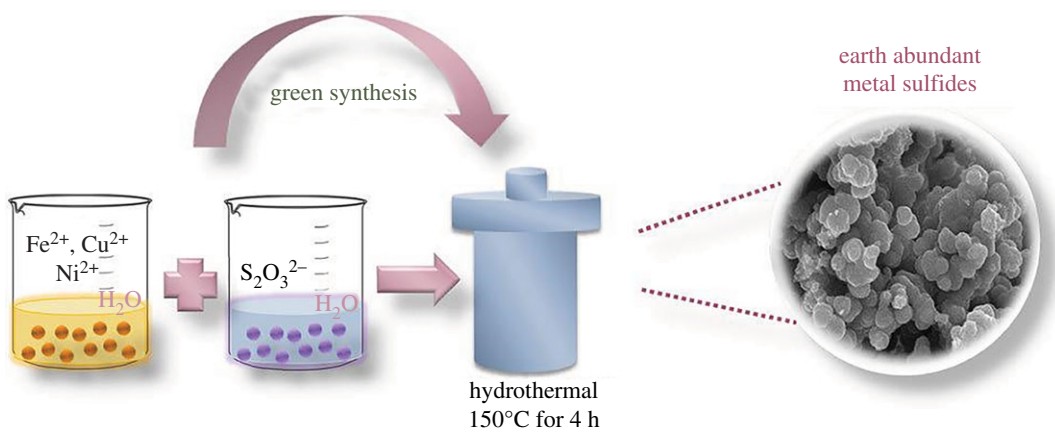

**Figure 20.** Schematic illustration of the hydrothermal synthesis of FeS$_2$, CuS and NiS$_2$ powders. Adapted from Springer.

CuS, NiS$_2$ and FeS$_2$, promising photocatalysts for the H$_2$ evolution reaction and degradation of dyes, among other applications, were hydrothermally prepared from Cu(NO$_3$)$_2 \cdot$ 3H$_2$O, Ni(NO$_3$)$_2 \cdot$ 6H$_2$O and Fe(NO$_3$)$_3 \cdot$ 9H$_2$O, respectively, and sodium thiosulfate as precursors (figure 20) [175]. Among these sulfides, FeS$_2$ showed the best activity in indigo carmine degradation (88%) and HER activity (32 µmol g$^{-1}$ h$^{-1}$). In the synthesis of metal sulfides, the hydrothermal synthesis is short (up to 4 h) and it can be used at low temperatures (150°C) without pH control and complexing agents, resulting in pure final products with high yield (greater than 93%). Thus, obtained metal sulfides are potential candidates for deeper study of water-splitting processes. In addition, spherical, uniform and highly dispersed semiconductor CdSe nanoparticles, stabilized with L-cysteine, were prepared in water at room temperature from elemental Se, Na$_2$SeO$_3 \cdot$ 5H$_2$O and CdCl$_2 \cdot$ 2H$_2$O as precursors [176]. L-Cysteine was found to play a role of a capping agent.

# 9. Materials, nanomaterials, composites and hybrids

A lot of investigations have been carried out, applying green chemistry routes, where possible, to synthesize various hybrids [177,178], materials and composites. The greener chemistry methods have

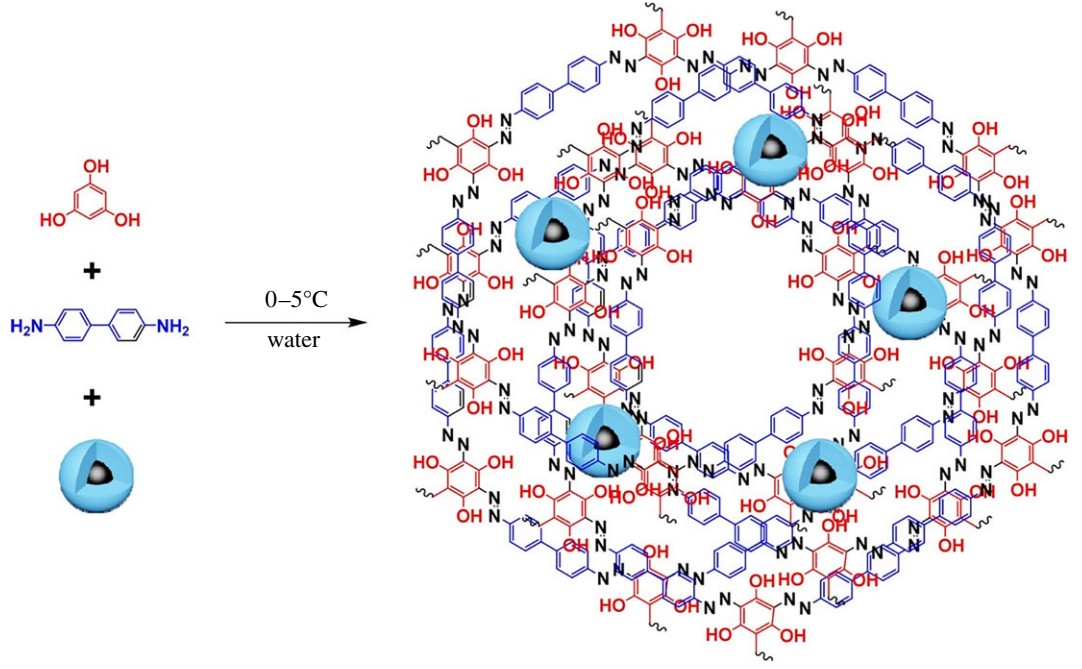

**Figure 21.** Synthesis of magnetic polymer hybrids. Adapted from ACS Publications.

been developed, in particular for titanium and manganese oxides, metal and metal oxide nanoparticles (see also sections above), mainly by hydrothermal and microwave-assisted techniques, as well as reflux [179]. Apart, several important green procedures have been offered for important types of materials as polymers and their composites, ceramics and bioceramics, cement, adsorbents and carbon-based aerogels. In the case of polymers, all 12 principles of green chemistry can be applied specifically to their fabrication [180], in particular, using agricultural products or wastes as renewable feedstocks instead of coal, natural gas or petroleum, water instead of harmful solvents, at normal pressure and room temperature, as well as other conditions with minimum pollution and risk of accidents. For the greener synthesis of polymers, the following substances can be used as renewable starting materials: cardanol, renewable plant oils, itaconic anhydride and lactic acid, meanwhile natural lipase enzyme can be used as catalyst in the polymerization processes [181].

As an example of a *water*-assisted procedure, the porous, stable, hydrophobic and high-surface-area polymer hybrids, possessing a host of phenolic hydroxyl groups, were prepared from *m*-trihydroxybenzene and 4,4′-diaminobiphenyl as precursors in water via azo coupling (figure 21), encapsulating magnetic $Fe_3O_4@SiO_2$ nanoparticles in polymer matrix, and further freeze-drying [182]. In respect of methylene blue as a model contaminant, this composite, due to its extensive conjugated system, revealed high adsorption capacity (1153 mg g$^{-1}$), easy magnetic recovery from water and minimum five times of reuse maintaining the same adsorption properties.

Particles of Au and Ag@Au nanocomposites were prepared in water at room temperature using core–shell microgels based on poly(*N*-isopropylacrylamide)/polyethyleneimine (PNIPAm/PEI) [183]. These microgels possess a role of templates and reductants, when Au(III) ion in $HAuCl_4·3H_2O$ is first reduced resulting in Au$^0$ nanoparticles, which are stabilized by PEI shells and then act as seeds for the formation of bimetallic nanoparticles. The formed nanocomposite was used as catalyst for *p*-nitrophenol reduction resulting in *p*-aminophenol, revealing their 25-fold higher performance compared with simple Au nanoparticles. Related stable raspberry-like polymer colloids, decorated with Ag nanoparticles (up to tens of nanometres in size, depending on metal precursor concentration, quantity of polymer and reaction time) in a raspberry-like fashion (figure 22), were obtained in EtOH solution by the interaction of polymer spheres and silver nitrate without additional stabilizers, showing good antibacterial properties [184].

Fabrication of emerging polymers made with or *containing elemental sulfur* is an important issue in respect of waste re-elaboration contributing to sustainability, since sulfur is a side product in the oil industry [185]. Special attention is paid to the polymerization reactions of renewable monomers (i.e. triglycerides and terpenes) with sulfur, which can be processed without solvent use, according to the green chemistry principles; polysulfide products applied for energy storage, photocatalytic water

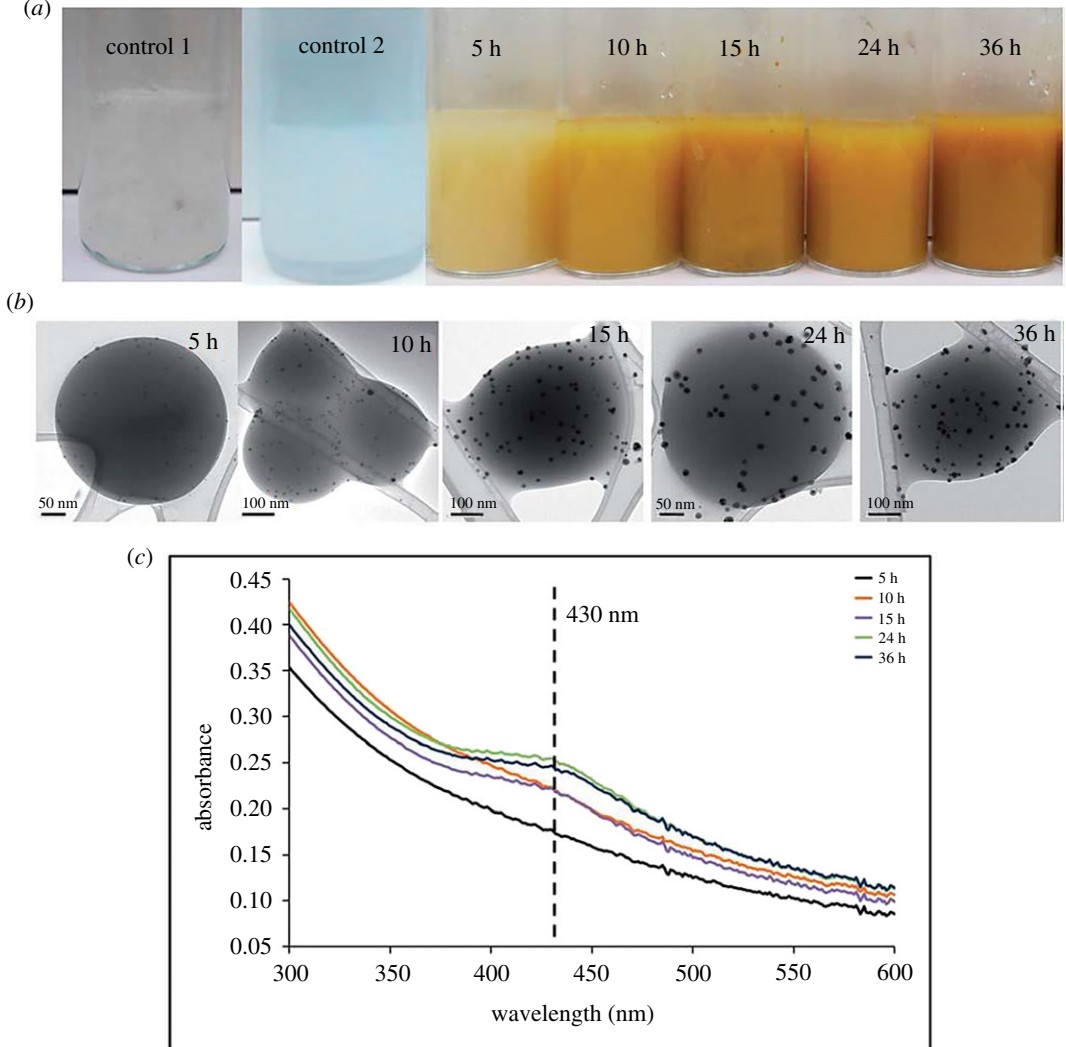

**Figure 22.** (*a*) Photographs of PGMA@AgNPs composite sphere solutions as a function of reaction time. Control 1: the product of AgNO$_3$ reacting with ethanol; control 2: the product of AgNO$_3$ reacting with purified PGMA spheres in DI water; (*b*) TEM images of PGMA@AgNPs composite spheres indicating the size evolution of AgNPs. Adapted from Royal Society of Chemistry.

splitting and remediation of heavy metals. These processes can be carried out by low-temperature polymerization (inverse vulcanization) in molten sulfur with control of the structure of formed S-containing polymers. One of the approaches is related with possible interactions in water as green solvent, involving inorganic polysulfides such as NaS-[S]$_n$-SNa and their interaction with polymer particles suspended in water as dispersion. The resulting polymers can contain up to 75% sulfur. Another option is the use of SC CO$_2$ in these processes. Meanwhile, S$_8$ is non-toxic, the lack of knowledge on polysulfides toxicity should also be taken into account, as well as necessary investigations on their recyclability, scalability and depolymerization back to monomers. At the same time, such sulfur-containing polymers can be biodegradable, since the S–S bond is subjected to photolysis and reduction.

Highly flexible polyurethane (PU) soft foams from Kraft lignin by its liquefaction in microwave-assisted conditions, using castor oil and polypropylene glycol triol as chain extenders [186]. Other possible applications of this material can be the decrease of glass transition temperature, packaging of furniture and car seats. PU chemistry can be developed by other green processes, for instance, the catalytic reaction of CO$_2$ with epoxidized soya bean oil as a starting oligomer in a high-pressure reactor (140°C, CO$_2$ pressure of 1 MPa, 800 r.p.m.) leads to triglycerides of cyclocarbonate- and epoxide-containing derivatives of carboxylic acids [187]. Also, piperazine-containing bis-phenol formaldehyde polymer was fabricated by a microwave-assisted method [188]. It was revealed that this polymer inhibits steel corrosion in HCl by adsorption mechanism. With a similar anti-corrosion purpose and also for creation of a self-healing coating, the nanocapsules with a natural component,

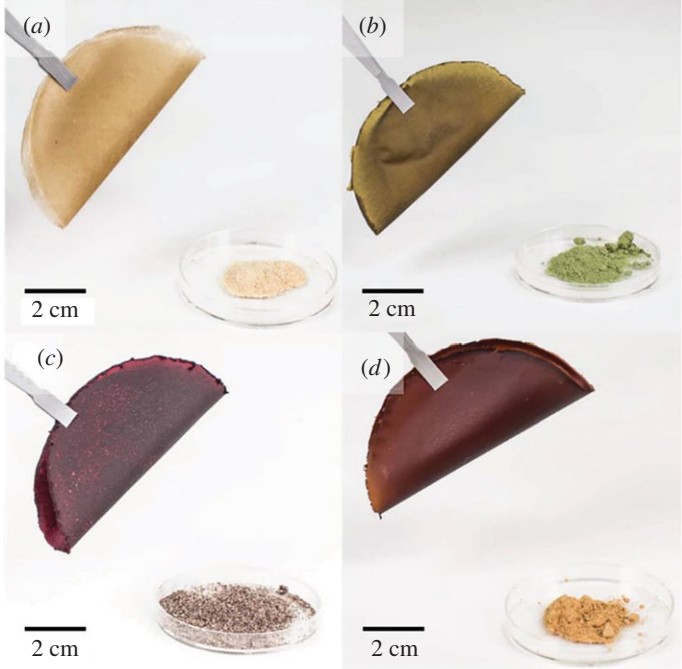

**Figure 23.** Images of obtained bioplastic films and the original vegetable powder used in the process: (*a*) carrot bioplastic, (*b*) parsley bioplastic, (*c*) radicchio bioplastic and (*d*) cauliflower bioplastic. Adapted from Royal Society of Chemistry.

inhibiting corrosion, were prepared by encapsulation of *Az. indica* in the polymeric shell of ureaformaldehyde under sonication and further polymerization *in situ* [189]. Green aspects were also reported for processes of fabrication of polyvinylchloride (PVC) [190] and several graft copolymers based on acrylic monomers and starch [191].

Certain attention is paid to *biodiesel*, *bioplastics*, *films* on their basis and *polymer biocomposites*. Thus, wood pulp was mechanically disintegrated and oxidized using magnetic Fe@MagTEMPO catalyst (30 nm in size; TEMPO = 4-oxo-2,2,6,6 tetramethylpiperidine-1-oxyl; $Fe_3O_4$ as a magnetic component; NaClO as oxidant and NaBr as promoter) in water under sonication resulting in nanofibrillated cellulose (fibre size 5 nm) [192]. The role of TEMPO is the diffusion into the fibre walls and oxidation of wood pulp, converting primary hydroxyls into aldehydes and carboxylic acids. The catalyst can easily be recycled and NaClO is transformed to environmentally friendly NaCl, corresponding to the green chemistry principles. Several vegetable waste materials (cauliflower, parsley, carrot and radicchio), without any pre-processing, were converted into biodegradable bioplastic films (figure 23) in aqueous 5% HCl in one-step process at r.t. [193]. These films consist of fused cellulose crystals, containing soluble ingredients such as sugars and pectin. Mechanical properties of the formed films, completely made of wastes of the vegetables above, are similar to those of starch bioplastics. Antioxidant properties of the original vegetables are conserved in the films due to mild synthesis conditions; the migration into food is low (less than $10 \ \mathrm{mg \ dm^{-2}}$). In addition, it is possible to combine these bioplastics with a variety of other polymers, resulting in new composites, for instance, polyvinyl acetate (PVA)-carrot bioplastic, suitable for packaging purposes.

Some biocomposites were obtained from recycled bioplastic polylactic acid and treated cellulosic fibres applying MW-irradiation technique combined with enzymatic treatment [194]. This method seems to be very reliable for biocomposite reinforcement, showing an increase in Young's modulus and tensile strength. Also, chitosan cross-linked Ag-containing nanocomposites were compared with Ag nanoparticles obtained with the aid of garden rhubarb (*Rheum rhabarbarum*) [195,196] in respect of toxicity against HeLa cell line, showing higher toxicity in the last case. Related hyaluronan fibres containing Ag nanoparticles were prepared (figure 24) from alkali solution (NaOH) of natural polysaccharide hyaluronic acid via wet-spinning method (nozzle diameter 0.4 mm), further complexation with $Ag^+$ ions, their reduction and stabilization by the material of fibres under stirring at 80°C [197]. $Ag^0$ nanoparticles were found to be stable for 90 days without aggregation. This composite material can be applied in dressing purposes, among other possible applications. In addition, *biodiesel* (methyl or ethyl ester of vegetable oil) was fabricated from several vegetable oils

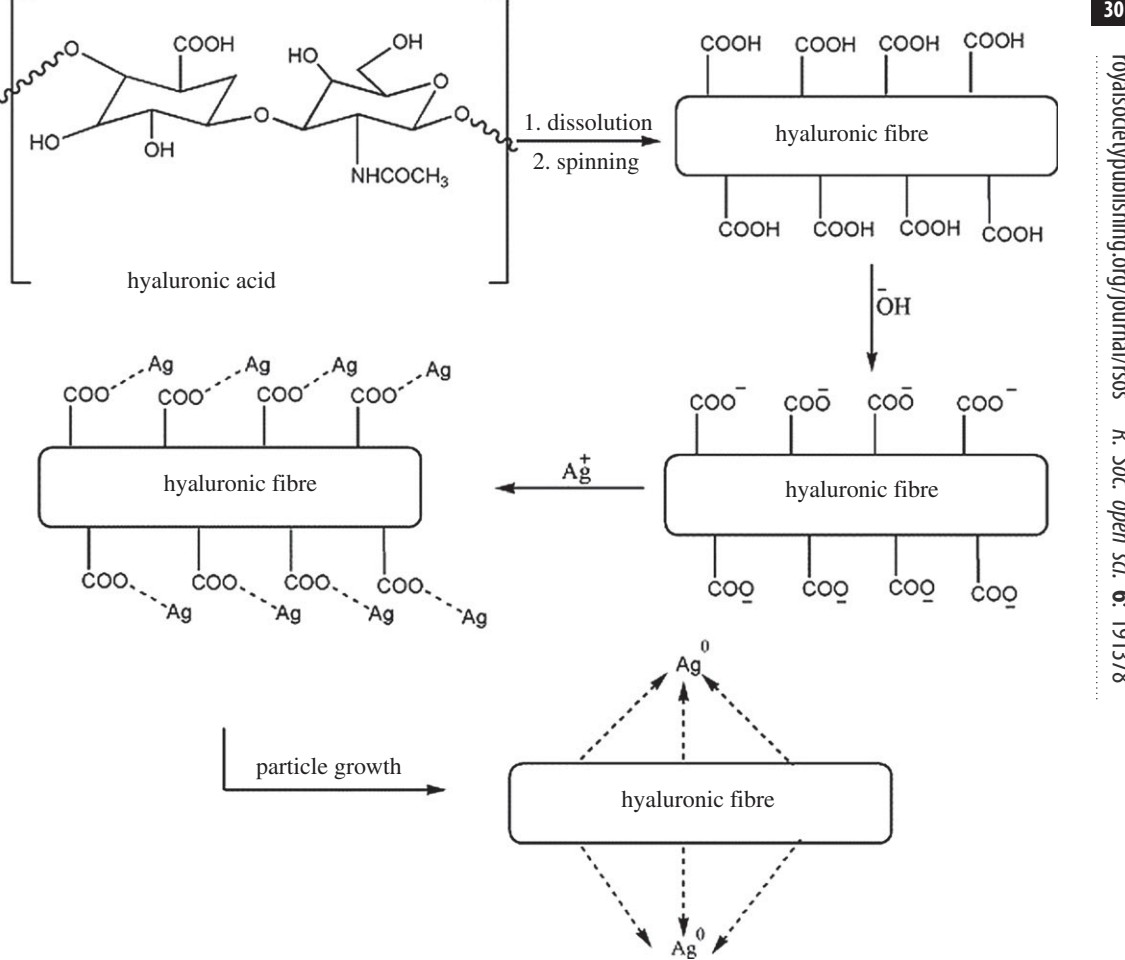

**Figure 24.** Preparation of hyaluronan fibre with silver nanoparticles.

such as jawas oil, groundnut oil, soya bean oil, sesame oil, mustard oil and coconut oil [198]. The reaction steps included the interaction of NaOH with methanol yielding $CH_3ONa$, its further stirring with groundnut oil or other oils, and separation of two layers (glycerol and biodiesel). The yield of biodiesel was found to be higher in the case of groundnut, soya bean and jawas oil.

The area of *adsorbents* applies green chemistry processes too. It is known that the activated carbon is mostly used for dye removal, but its main disadvantage is high cost. So, cheaper adsorbents are under search and development. As green methods for the fabrication of adsorbents for dye removal from wastewater, the ultrasonication, microwave-assisted processes and use of ILs are considered as suitable to satisfy the 12 rules of green chemistry. In addition, a variety of functionalization (physical, chemical and biological) methods is known to improve their efficiency, providing faster adsorption interactions [199]. Resulting composites can contain minerals, nanostructurized metals and other nanomaterials, magnetic additives, polymers, carbon allotropes (i.e. graphene), cellulose, but they have not yet been scaled up above some kilograms. Among other important modern trends for dye sequestration by adsorbents, the development of greener adsorbents includes the use of biobased feedstock, application of non-hazardous functionalization techniques and hydrothermal treatment, creating abundant O-containing functional groups, control of biofilm thickness and pore size for better dye uptake, etc.

An example of green material, prepared from natural product, is the graphenic adsorbent, fabricated from palm oil mill effluent, which was found to be highly efficient for dye-polluted wastewater purification [200]. Also, iron-containing nano-impregnated adsorbent was prepared from ferrous sulfate as metal source, using black tea solution, heated at 80°C for 1 h and further sonication with 1-butyl-3-methylimidazolidium bromide solution for 24 h. Also, iron-containing nano-impregnated adsorbent was prepared from ferrous sulfate as metal source, using black tea solution, heated at 80°C for 1 h and further sonication with 1-butyl-3-methylimidazolidium bromide solution for 24 h [201]. The formed material was applied for selective and fast fluoride removal from water (up to 90%, figure 25), whose mechanism is based on the difference in electronegativity of $F^-$ and $Br^-$ and includes interaction of its positive charges with fluoride anions

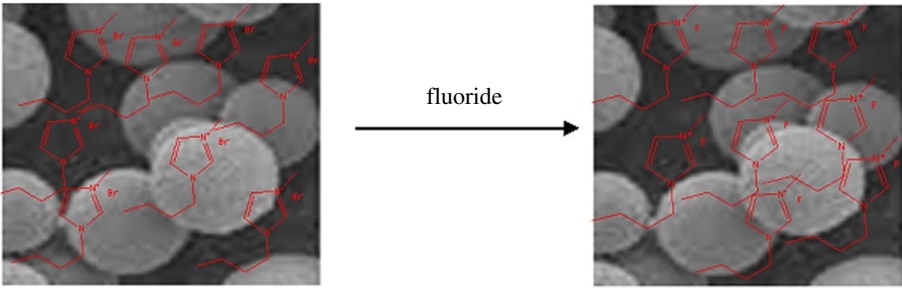

**Figure 25.** Mechanism of fluoride adsorption. Adapted from Elsevier.

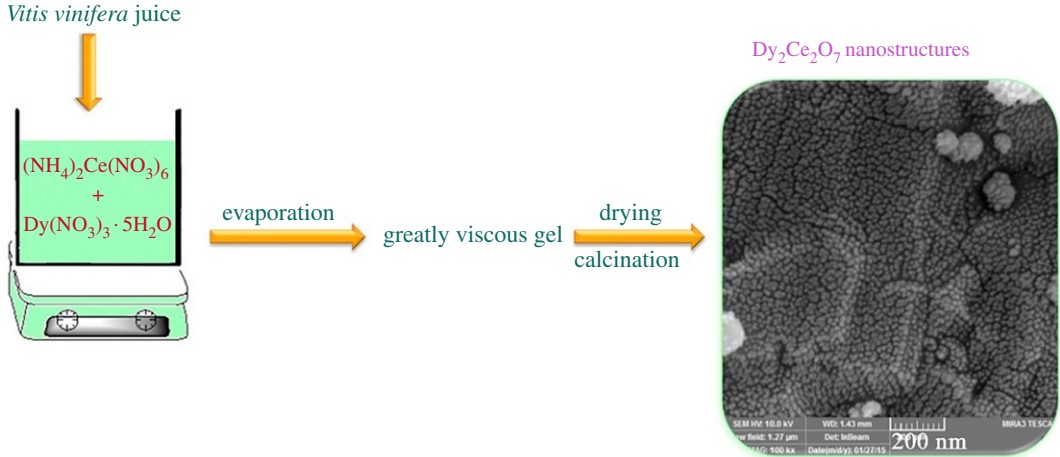

**Figure 26.** Schematic diagram of the production of $Dy_2Ce_2O_7$ nanostructures. Adapted from Elsevier.

replacing bromide anions. This adsorption process is exothermic, scalable and can be applied at natural water pHs. In addition, magnetic $Fe_3O_4$/PPw (PPw, papaya peel waste) bio-adsorbent, synthesized in one step from $FeCl_2 \cdot 4H_2O$ and $FeCl_3 \cdot 6H_2O$ as metal source by slow growth of $Fe_3O_4$ nanoparticles on PPw, was used for $Pb^{2+}$ extraction from wastewater, allowing excellent lead determination and easy separation by an external magnet [202]. Other adsorbents in aerogel form will be shown below.

Several *ceramic and bioceramic* materials have been prepared by green techniques. Thus, ceramic powders were fabricated [203] by green mechanochemical method (co-milling using 10 mm steel balls in steel vessels at 1100 r.p.m. for 2–5 h), applied to three systems: SiC and $SiC/TiO_2$ (6 : 1 and 3 : 1 ratios). Quality of products (crystallinity and shape of microparticles) was found to be dependent on $TiO_2$ content, being better in the case of its small amounts; 2 h time seems sufficient for SiC coating with $TiO_2$ without considerable further morphology change. $Dy_2Ce_2O_7$ ceramic nanostructures were prepared (figure 26) from dysprosium nitrate and ceric ammonium nitrate using *Vitis vinifera* juice, containing fructose and glucose as reductants and capping agents, as green fuel and further calcination at 400°C for 5 h [204]. The grain size, shape and photocatalytic properties of $Dy_2Ce_2O_7$ in degradation of pollution such as rhodamine B, methyl orange and 2-naphthol were found to be depended on calcination temperature. Their destruction mechanism is shown in reactions (9.1)–(9.4). In addition, calcium silicate powders, suitable for biomedical applications, were hydrothermally obtained in a green medium from $Ca(NO_3)_2$ and TEOS ($Si(OC_2H_5)_4$) at three molar ratios (Ca/Si = 6 : 4, 5 : 5 and 4 : 6) and further sinterization at 600–1000°C for 2 h, comparing this process with classic sol–gel preparation of this material [205]. The hydrothermal process excludes organic solvents and acids, leading to different powders depending on the initial Ca : Si precursor ratio and calcination temperature. Green-method-synthesized optical $Yb:CaF_2$ ceramics are also known [206].

$$Dy_2Ce_2O_7 \text{ sample} + h\nu \rightarrow Dy_2Ce_2O_7 \text{ sample}^* + e^- + h^+, \tag{9.1}$$

$$e^- + O_2 \rightarrow O_2^{-\bullet}, \tag{9.2}$$

$$h^+ + H_2O \rightarrow H^+ + OH^\bullet \tag{9.3}$$

and $\quad$ undesirable contaminants $+ \quad OH_\bullet$ or $O_2^{-\bullet} \rightarrow$ degradation products. $\tag{9.4}$

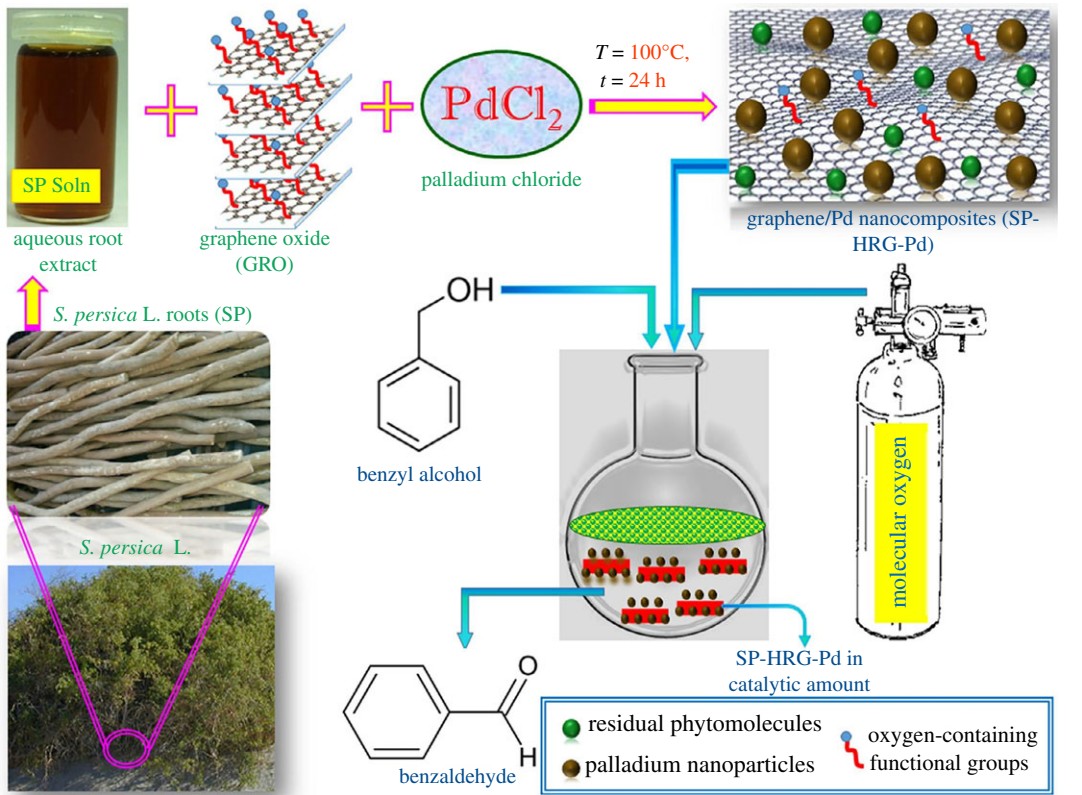

**Figure 27.** Schematic illustration of the green synthesis of Pd@graphene (SP-HRG-Pd) nanocomposites using *Salvadora persica* L. root extract (RE) as bioreductant. The as-prepared nanocomposite is also applied as oxidation catalyst for the transformation of various alcohols. Adapted from Elsevier.

Because of enormous scale of *cement* production worldwide as a key construction material, huge amounts of $CO_2$ emissions are one of the strongest negative impacts of cement industries. Therefore, greener techniques for cement fabrication (in particular, ordinary Portland cement, OPC) and search of replacement materials (i.e. silica fumes, blast furnace slag or fly ash) are in permanent development, looking for new production methods, novel formulations and use of alternative fuels with low carbon content, such as biomass [207]. Also, three alternative cements are promising in respect of greenhouse gas emissions and other characteristics: magnesia cements based on MgO key ingredient, sulfoaluminate cements with high alumina content, blended OPC-based cements (homogeneous mixture of a replacement material and OPC) and alkali-activated cements with a considerable amount of aluminosilicate phase [208]. A host of factors need to be taken into account in green chemistry for cement design: composition (cations of transition or alkali metals, pH, metal oxides, oxy anions, etc.), structural characteristics (amorphous phase versus crystalline phase, two-dimensional versus three-dimensional) and properties (solubility, durability, chemical stability, morphology, hydration properties, strength, etc.). Novel greener methods must be adapted to local raw materials in different countries and regions, resulting in the product with the same or very similar properties and characteristics.

*Graphene*-based nanocomposites [209], hybrids and gels represent a relatively new and important research area, with catalytic, energy and environmental applications, so their possible greener fabrication and possible scaling-up is promising field. Thus, palladium(Pd)@graphene nanocomposites (SP-HRG-Pd, figure 27) were fabricated by the simultaneous reduction of $PdCl_2$ and GO with aid of *Salvadora persica* L. root extract (RE) [210]. Metal reduction is favourable due to the action of terpenoids and polyphenolic (flavonoids) contents in this plant, also contributing to the uniform binding of palladium nanoparticles on graphene surface and overall stabilization of whole surface of the formed nanocomposite. The hybrid reveals properties such as water dispersibility and catalytic activity in the processes of selective aromatic alcohol oxidations. Three-dimensional graphene hydrogel (GH) with macroporous frameworks, hydrothermally prepared (figure 28) in an ammonia-glucose system via self-assembling with further freeze-drying, was immersed in $KMnO_4$ under heating, resulting in the δ-$MnO_2$/GH composite (figure 29), applied in a supercapacitor with good characteristics [211]. In addition, MWCNT-graphene aerogels, prepared from hydrogel precursors with

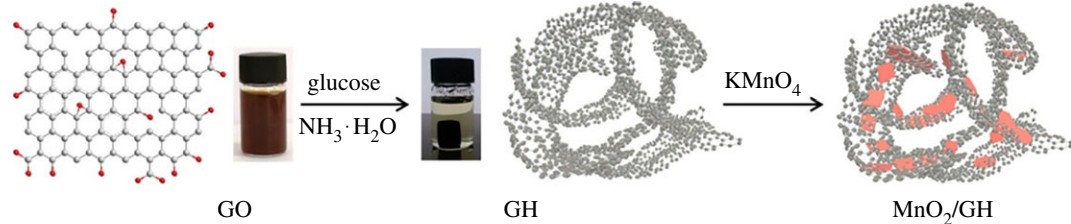

**Figure 28.** Schematics of the synthesis process of GH and MnO$_2$/GH. Adapted from ACS Publications.

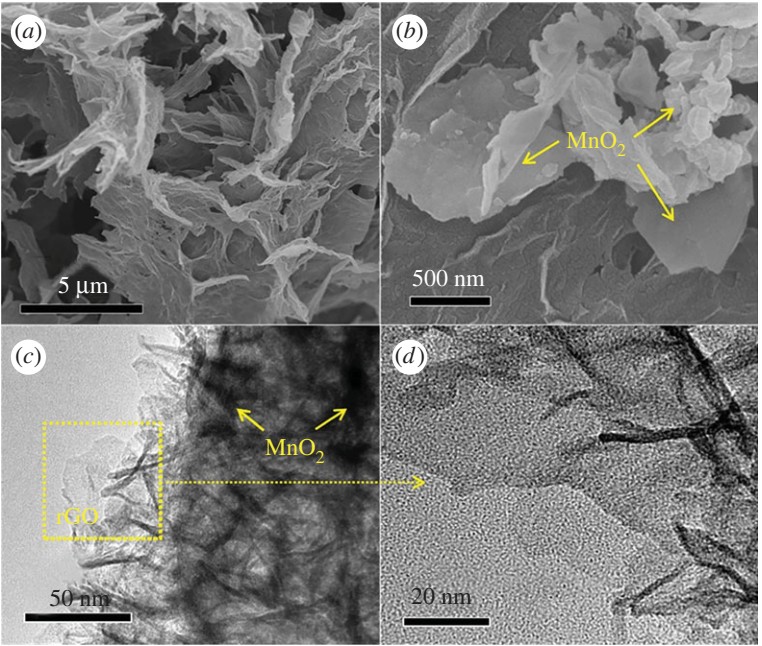

**Figure 29.** SEM images of GH (*a*) and MnO$_2$/GH (*b*) and TEM images of the MnO$_2$/GH sample (*c,d*). Adapted from ACS Publications.

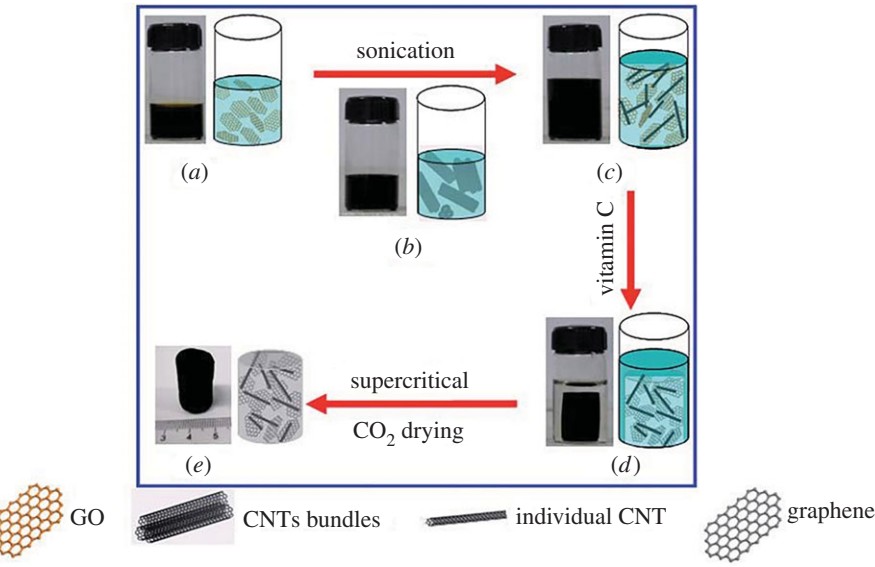

**Figure 30.** Schematic diagram for green synthesis of the graphene–CNT hybrid aerogels: (*a*) pre-dispersed GO solution; (*b*) pre-dispersed CNT suspension; (*c*) the mixture of GO and CNT after sonication; (*d*) the graphene–CNT hybrid hydrogel and (*e*) the graphene–CNT hybrid aerogel. In each subdivision graph, the left part is the digital photo of the real substance and the right part is the drawing photo of the imaginary substance. The CNTs employed here are directly applicable to pristine (MWCNTs) or acid-treated (c-MWCNTs) multi-wall CNTs. Adapted from Royal Society of Chemistry.

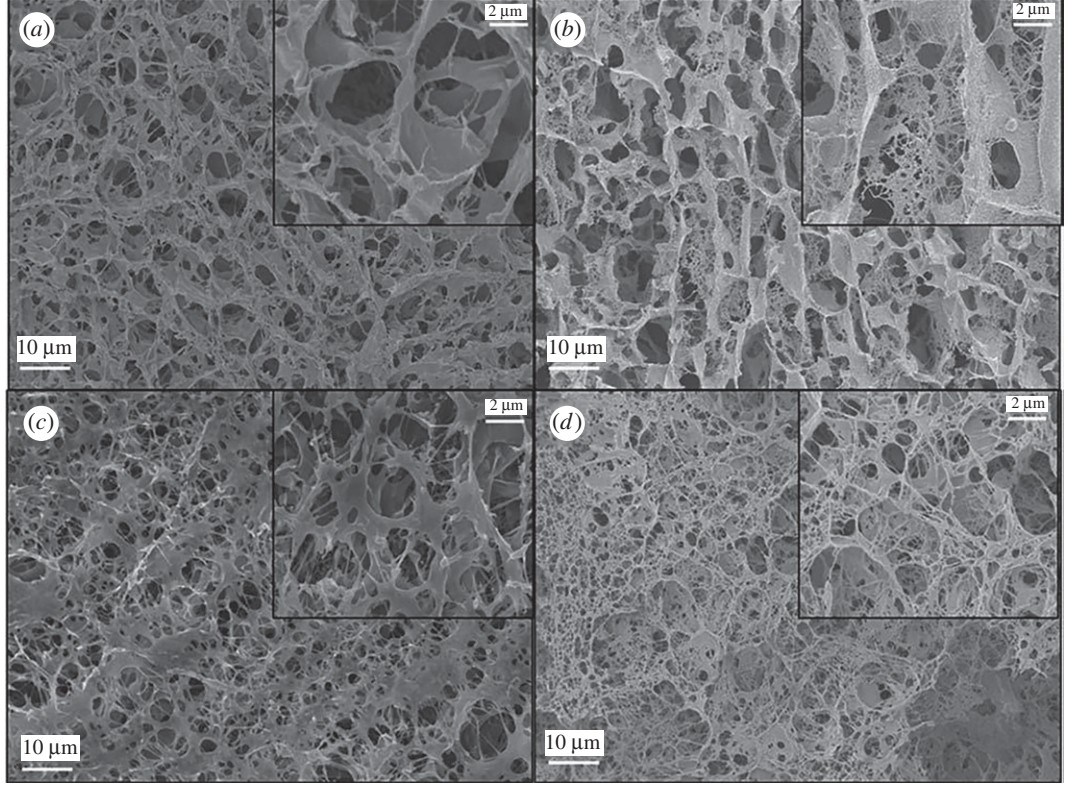

**Figure 31.** SEM images of the bottom surface of the aerogels: (*a*) uncoated PVA aerogel, (*b*) coated PVA aerogel, (*c*) uncoated PVA/CNF aerogel and (*d*) coated PVA/CNF aerogel. Adapted from Royal Society of Chemistry.

Vitamin C by SC $CO_2$ drying (figure 30) [212], showed excellent characteristics in elimination of organic dyes and deionization of light metal salts due to large Brunauer–Emmett–Teller area, light weight and hierarchically porous structure. Several heavy metals ($Ag^+$, $Pb^{2+}$) can be also successfully adsorbed. Finally, both superhydrophobic and superoleophilic organic aerogels (figure 31) on the basis of cross-linked PVA/CNF (polyvinyl alcohol and cellulose nanofibril, respectively) were obtained through green freeze-drying process and further silanization by CVD treatment with methyltrichlorosilane [213]. The products, having ultra-low densities (less than $15 \, kg \, m^{-3}$), are excellent adsorbents for crude oil and other types of oils (capturing 44–96 times of their own dry weight) and heavy metals ($Hg^{2+}$, $Pb^{2+}$), thus being useful for water cleaning.

# 10. Conclusion and further outlook

In twentieth to twenty-first centuries, due to enormous growth of chemical, metallurgical, pharmaceutical and other industries, the problems of environment, related with pollution, have become threatening to the further development of humanity. In the middle of the twentieth century, not much attention was paid to the ecological problems worldwide. Only in the second half of the last century, several intents to improve this situation have started at the international level. In this respect, green chemistry is a considerable contribution to this field. The 12 key principles of green chemistry, on whose basis the present review has been written, mean, in particular, the use of alternative raw materials (such as plants, agricultural wastes), renewable sources and effective catalysts, non-hazardous manufacture when possible without the use of toxic chemicals and ideal case when $CO_2$ and water as by-products, prevention of unnecessary wastes, lower-hazard chemical reactions, minimization of energy consume, leading to degradable reaction products, non-persisting in the environment. Energy-efficient processes such as biotransformations, photochemistry, ultrasound or microwaves and use of bioactive molecules in chemical processes yielding high-value molecules—all these are green processes.

The green chemistry methods include several non-contaminating physical methods as microwave heating, ultrasound-assisted and hydrothermal processes or ball milling, frequently in combination with the use of natural precursors, which are of major importance in the greener synthesis, as well as

solvent-less and biosynthesis techniques. Biological methods (the use of bacteria, viruses, yeasts, plant extracts, fungi and algae) perfectly fit to the green chemistry, in particular to nanochemistry, resulting in biologically produced nanoparticles, which are non-toxic, stable, environmentally friendly and cost effective. Plant extracts contain polyphenols, terpenoids, proteins, enzymes, peptides, sugars, phenolic acids and bioactive alkaloids as a driving force for nanoparticle formation.

It is important to highlight that green chemistry and the awareness that this has given to the scientists has allowed us to make exhaustive revisions of the traditional methods to optimize them and to implement them based on the standards of the present time and, in some cases, the traditional methods have been substituted, bringing as a consequence more optimized processes both at the laboratory level and at the industrial level, as it has been seen in the reports mentioned in this review.

The processes and syntheses that have as main focus the green chemistry that have been found in the literature has great advantages over traditional methods, among which are the reduction of toxic emissions to the environment, reactions that do not generate by-products and processes more optimal, which in turn reduce production costs, such as costs derived from the treatment or storage of waste and the adverse health effects on personnel working in the process.

In organic chemistry, the green chemistry reactions include several processes in the formation of C–C bonds (i.e. Suzuki and Glaser coupling, Knoevenagel condensation, McLurry, Wittig, Gewald, Michael, Reformatsky and Gringard reactions, arylaminomethylation, etc.), C–N bonds (synthesis of oximes, imines, azines, guanidines, (thio)semicarbazones, nitrones, N-arylation of amines, etc.), C–O, C–S, C-Hal, C–H and other bonds, cycloaddition reactions, reductions and oxidations, as well as for functionalization of nanocarbons (SWCNTs, graphene and $C_{60}$) and in supramolecular chemistry. A range of nanosized materials and composites can be produced by greener routes, including nanoparticles of metals, non-metals, their oxides and salts, aerogels or QDs. At the same time, such classic materials can be improved or obtained by cleaner processes as cement, ceramics and bioceramics, adsorbents, polymers, bioplastics, biodiesel and biocomposites.

All methods from the above discussed set of greener laboratory techniques cannot be always considered as indeed green upon scaling up. As an example, ball milling in a larger scale could be a source of solid microsize pollution. Microwave heating, hydro/solvothermal reactions and ultrasound treatment, being scaled to kilograms and more, could consume too much electricity, which needs to be produced by non-green methods, such as, for instance, by combustion of carbon. In other words, upon scaling up, the principal sense of greener synthesis could be lost, transforming it to a common physico-chemical process. For that matter, biological methods can be much more economical and profitable (although slower) than chemical ones. Among the discussed types of materials and compounds, synthesized by greener methods only, several organic compounds and metal nanoparticles are commercially available, in a difference of many other mentioned chemicals. For some nanomaterials and chemical compounds, greener methods are not economically profitable. For example, co-authors of this review accidentally discovered some years ago a greener method of fabrication of carbon nano-onions at 40°C (instead of their standard high-temperature route) in a laboratory ultrasonic equipment using commercially available MWCNTs and a water-soluble substituted cobalt(II) phthalocyanine 'theraphthal' (TP) [214]. However, this technique is not economically profitable because of high cost of the TP. In addition, some methods, although being obviously green, are less common (rare) even at laboratory scale, for instance, magnetic field-assisted synthesis; other ones like MW heating are now common and almost classic techniques. All this, together with economic factors, influence the necessity and real availability of scaling up laboratory-level green procedures.

Currently, courses on green chemistry are offered for students in several universities worldwide. Also, green chemistry congresses of different levels take place every year. In addition, the ideas of green chemistry have been converted in obligatory rules in some departments of chemistry, for instance, in several Mexican universities including the Autonomous University of Nuevo Leon in Monterrey. This city is known as an industrial capital of Mexico, having a lot of chemical/metallurgical industries and, as a consequence, much contamination. Obviously, hypothetical correct application of the classic 12 rules of green chemistry in industrial processes could considerably improve the ecological situation in contaminated regions worldwide. Currently, many of the green chemistry reactions described above have been carried out at the laboratory scale. Therefore, the development of scaled-up greener processes is recommended first of all, despite obvious economic problems and lesser competitiveness for companies introducing these routes.

Data accessibility. This article does not contain any additional data.

Authors' contributions. O.V.K. carried out the extensive search of literature data and re-elaboration of data. B.I.K. offered the idea to write this large review and carried out the literature search and re-elaboration of data. C.M.O.G. collected

much literature data and carried out most technical work on them. Y.P.M. and I.L. carried out a considerable work on the literature search and discussions during the preparation of manuscript. All authors gave final approval for publication and agree to be held accountable for the work performed therein.

Competing interests. We declare we have no competing interests.

Funding. C.M.O.G. is grateful to the Conacyt-Mexico for his PhD scholarship.

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
