## [Reviewer comments · Royal Society Open Science]

Review History

Decision letter (RSOS-191261.R0)

05-Aug-2019

Dear Dr Kharissov:

Manuscript ID: RSOS-191261

Title: "Greener synthesis of compounds, materials, hybrids, nanomaterials, and nanocomposites: state of the art and modern trends."

Thank you for submitting the above manuscript to Royal Society Open Science. Your paper was sent to reviewers and their comments are included at the bottom of this letter.

In view of the concerns raised by the reviewers, the manuscript has been rejected in its current form. However, a new manuscript may be submitted which takes into consideration these comments.

Please note that resubmitting your manuscript does not guarantee eventual acceptance, and that your resubmission will be subject to peer review before a decision is made.

Your resubmitted manuscript should be submitted by 02-Feb-2020. If you are unable to submit by this date please contact the Editorial Office.

Yours sincerely,
Dr Ellis Wilde
Publishing Editor, Journals

On behalf of the Subject Editor Professor Anthony Stace and the Associate Editor Dr Andrew Harned

REVIEWER REPORTS:

Associate Editor Comments to Author:

RSC Associate Editor

Comments to the Author:

My initial reaction is that in its current form, the review is too long for our journal. I also felt that the scope is a bit too broad and, consequently, the manuscript as a whole was a bit unorganized. Nevertheless, I think this topic would be of use to our readership. I suggest the authors submit a revised manuscript that takes into account the following suggestions and comments.

(1) There has been a lot of coverage in the literature on greener methods for small molecule synthesis. As a result, I am less excited by another review on this topic. In contrast, a solid focus on greener synthesis of materials is highly topical. I strongly suggest that coverage of green synthesis of small molecules (Section 3) be removed altogether, or at the very least, pared back substantially.

(2) I very much like the coverage of the different methods (Section 2), but I wonder if the discussion can be made more concise?

(3) All structures should be checked for accuracy. There were several instances where it looked like double bonds were missing, or other parts of the structures were incorrect.

(4) Wherever possible, the green methods discussed in the review should be compared against "more traditional" methods. What makes existing methods less desirable or environmentally

unfriendly? (High cost? High energy consumption? Toxic reagents?) The authors should avoid making this review a simple catalog of methods.

Reviewers' Comments to Author:

RSOS-191378.R0

Review form: Reviewer 1

Is the manuscript scientifically sound in its present form?

Yes

Are the interpretations and conclusions justified by the results?

Yes

Is the language acceptable?

Yes

Do you have any ethical concerns with this paper?

No

Have you any concerns about statistical analyses in this paper?

No

Recommendation?

Accept with minor revision (please list in comments)

Comments to the Author(s)

Please, check the text thoroughly, as there are some minor grammatical mistakes, although the text is readable and mostly clear.

Review form: Reviewer 2

Is the manuscript scientifically sound in its present form?

Yes

Are the interpretations and conclusions justified by the results?

Yes

Is the language acceptable?

Yes

Do you have any ethical concerns with this paper?

No

Have you any concerns about statistical analyses in this paper?

No

Recommendation?

Accept as is

Comments to the Author(s)

Journal: Royal Society Open Science RSOS-191378

Reviewer's Comments

This manuscript has reported a complete and critical overview on several greener synthesis and fabrication of inorganic, organic and coordination compounds, materials, nanomaterials, hybrids, and nanocomposites. Details of greener techniques for synthesis of both well-known chemical compounds and completely new materials are also discussed. The manuscript is very exciting and topic is of scientific interest.

Authors have addressed and reviewed major physical methods as microwave heating, ultrasound-assisted and hydrothermal processes or ball milling utilized in the greener synthesis, as well as solventless and biosynthesis techniques. Then scaling up of green processes for profit are also explored. I think this manuscript is in very good shape and very important.

Overall, this paper is attractive from the standpoint of greener synthesis and fabrication research. The analysis of the topic is very significant and must be interesting to the readers of Royal Society Open Science. I strongly recommend it for publishing in Royal Society Open Science.

Review form: Reviewer 3

Is the manuscript scientifically sound in its present form?

No

Are the interpretations and conclusions justified by the results?

Yes

Is the language acceptable?

Yes

Do you have any ethical concerns with this paper?

No

Have you any concerns about statistical analyses in this paper?

No

Recommendation?

Accept with minor revision (please list in comments)

Comments to the Author(s)

This review article by Kharisov and co-authors on greener synthesis of various compounds and materials is timely and useful. It covers several sub-topics. I recommend the publication after some changes and improvements. Hope that the following observations, comments and suggestions are useful to the authors and editor.

Title covers several topics. Different categories however are somewhat ill defined. What is the difference between each category such as “compounds, materials, hybrids, nanomaterials, and nanocomposites” listed in the title. For example, most of them can be described as materials or molecules and materials. Authors should think more carefully about this and use a better title.

Methods of synthesis section is clear and nicely written. However, the type of items covered in sections 4-8 are less clear from reading first part of each section. It would be useful to describe what type of materials/molecules covered in each section in the first sentences of each subsection. Also it is useful to direct the reader to any recent reviews on such topics in each section.

Some editing at the editorial office needed to improve the language and presentation but overall it is good.

References should be improved significantly as they reflect on this review. There are inconsistencies, missing details, etc. Authors and journal office may want to check the list carefully -- See just a partial list below,

ref. 10, 17, 30, 48, 179, ..etc. have no titles or details apart from a web address.

Ref 73, 111, ...etc are incomplete

A few are from questionable and low-quality journals and non-peer reviewed items. Please use better options and improve

Decision letter (RSOS-191378.R0)

16-Sep-2019

Dear Dr Kharissov:

Title: Greener synthesis of compounds, materials, hybrids, nanomaterials, and nanocomposites: state of the art and modern trends.

Manuscript ID: RSOS-191378

Thank you for submitting the above manuscript to Royal Society Open Science. On behalf of the Editors and the Royal Society of Chemistry, I am pleased to inform you that your manuscript will be accepted for publication in Royal Society Open Science subject to minor revision in accordance with the referee suggestions. Please find the reviewers' comments at the end of this email.

The reviewers and handling editors have recommended publication, but also suggest some minor revisions to your manuscript. Therefore, I invite you to respond to the comments and revise your manuscript.

Please also include the following statements alongside the other end statements. As we cannot publish your manuscript without these end statements included, if you feel that a given heading is not relevant to your paper, please nevertheless include the heading and explicitly state that it is not relevant to your work. We have included a screenshot example of the end statements for reference.

- Acknowledgements

- Funding statement

Please include a funding section after your main text which lists the source of funding for each author.

Because the schedule for publication is very tight, it is a condition of publication that you submit the revised version of your manuscript before 25-Sep-2019. Please note that the revision deadline will expire at 00.00am on this date. If you do not think you will be able to meet this date please let me know immediately.

Best wishes,

Dr Laura Smith
Publishing Editor, Journals

On behalf of the Subject Editor Professor Anthony Stace and the Associate Editor Dr Andrew Harned.

RSC Associate Editor

Comments to the Author:

All referees have expressed enthusiasm for this review and the Editor believes this is a very topical subject that should draw attention from our readership. The referees all believe this to be a well written document, in general, but there are a few relatively minor aspects that need to be addressed before final acceptance. The authors should submit a revised manuscript based on the supplied reports. In particular, they should check all references to ensure they are complete and in the appropriate format for this journal. In addition, they should carefully consider the quality of the references used in this review. The Editor appreciates the thoroughness the authors have displayed, but also agrees with the reviewers that, when possible, higher quality references should be used when available and appropriate. If possible, try to limit the number of references to non-peer reviewed sources.

Reviewer comments to Author:

Reviewer: 1

Comments to the Author(s)

Please, check the text thoroughly, as there are some minor grammatical mistakes, although the text is readable and mostly clear.

Reviewer: 2

Comments to the Author(s)

Journal: Royal Society Open Science RSOS-191378

Reviewer's Comments

This manuscript has reported a complete and critical overview on several greener synthesis and fabrication of inorganic, organic and coordination compounds, materials, nanomaterials, hybrids,

and nanocomposites. Details of greener techniques for synthesis of both well-known chemical compounds and completely new materials are also discussed. The manuscript is very exciting and topic is of scientific interest.

Authors have addressed and reviewed major physical methods as microwave heating, ultrasound-assisted and hydrothermal processes or ball milling utilized in the greener synthesis, as well as solventless and biosynthesis techniques. Then scaling up of green processes for profit are also explored. I think this manuscript is in very good shape and very important.

Overall, this paper is attractive from the standpoint of greener synthesis and fabrication research. The analysis of the topic is very significant and must be interesting to the readers of Royal Society Open Science. I strongly recommend it for publishing in Royal Society Open Science.

Reviewer: 3

Comments to the Author(s)

This review article by Kharisov and co-authors on greener synthesis of various compounds and materials is timely and useful. It covers several sub-topics. I recommend the publication after some changes and improvements. Hope that the following observations, comments and suggestions are useful to the authors and editor.

Title covers several topics. Different categories however are somewhat ill defined. What is the difference between each category such as "compounds, materials, hybrids, nanomaterials, and nanocomposites" listed in the title. For example, most of them can be described as materials or molecules and materials. Authors should think more carefully about this and use a better title.

Methods of synthesis section is clear and nicely written. However, the type of items covered in sections 4-8 are less clear from reading first part of each section. It would be useful to describe what type of materials/ molecules covered in each section in the first sentences of each subsection. Also it is useful to direct the reader to any recent reviews on such topics in each section.

Some editing at the editorial office needed to improve the language and presentation but overall it is good.

References should be improved significantly as they reflect on this review. There are inconsistencies, missing details, etc. Authors and journal office may want to check the list carefully -- See just a partial list below, ref. 10, 17, 30, 48, 179, ..etc. have no titles or details apart from a web address.

Ref 73, 111, ...etc are incomplete

A few are from questionable and low-quality journals and non-peer reviewed items. Please use better options and improve

Author's Response to Decision Letter for (RSOS-191378.R0)

See Appendix A.

Decision letter (RSOS-191378.R1)

04-Oct-2019

Dear Dr Kharissov:

Title: Greener synthesis of chemical compounds and materials.
Manuscript ID: RSOS-191378.R1

It is a pleasure to accept your manuscript in its current form for publication in Royal Society Open Science. The chemistry content of Royal Society Open Science is published in collaboration with the Royal Society of Chemistry.

On behalf of the Subject Editor Professor Anthony Stace and the Associate Editor Dr Andrew Harned.

RSC Associate Editor

Comments to the Author:

The authors have incorporated the suggested changes offered by the reviewers. The new title is more concise, yet still conveys the topic at hand. This will be a welcome addition to the journal and I am pleased to recommend acceptance at this time.

Reviewer(s)' Comments to Author:

Appendix A

CORRECTIONS (please see below in red)

Dear Editor,

Many thanks for careful revision of our manuscript. Please see the list of changes below (marked in red)

Best regards,

Boris

16-Sep-2019

Dear Dr Kharissov:

Title: Greener synthesis of compounds, materials, hybrids, nanomaterials, and nanocomposites: state of the art and modern trends.

Manuscript ID: RSOS-191378

Thank you for submitting the above manuscript to Royal Society Open Science. On behalf of the Editors and the Royal Society of Chemistry, I am pleased to inform you that your manuscript will be accepted for publication in Royal Society Open Science subject to minor revision in accordance with the referee suggestions. Please find the reviewers' comments at the end of this email.

The reviewers and handling editors have recommended publication, but also suggest some minor revisions to your manuscript. Therefore, I invite you to respond to the comments and revise your manuscript.

Please also include the following statements alongside the other end statements. As we cannot publish your manuscript without these end statements included, if you feel that a given heading is not relevant to your paper, please nevertheless include the heading and explicitly state that it is not relevant to your work. We have included a screenshot example of the end statements for reference.

- Acknowledgements

- Funding statement

Please include a funding section after your main text which lists the source of funding for each author.

Reply: Statement added.

Because the schedule for publication is very tight, it is a condition of publication that you submit the revised version of your manuscript before 25-Sep-2019. Please note that the revision deadline will expire at 00.00am on this date. If you do not think you will be able to meet this date please let me know immediately.

Supplementary files will be published alongside the paper on the journal website and

posted on the online figshare repository (<https://figshare.com>). The heading and legend provided for each supplementary file during the submission process will be used to create the figshare page, so please ensure these are accurate and informative so that your files can be found in searches. Files on figshare will be made available approximately one week before the accompanying article so that the supplementary material can be attributed a unique DOI.

Best wishes,

Dr Laura Smith
Publishing Editor, Journals

On behalf of the Subject Editor Professor Anthony Stace and the Associate Editor Dr Andrew Harned.

RSC Associate Editor

Comments to the Author:

All referees have expressed enthusiasm for this review and the Editor believes this is a very topical subject that should draw attention from our readership. The referees all believe this to be a well written document, in general, but there are a few relatively minor aspects that need to be addressed before final acceptance. The authors should submit a revised manuscript based on the supplied reports. In particular, they should check all references to ensure they are complete and in the appropriate format for this journal. In addition, they should carefully consider the quality of the references used in this review. The Editor appreciates the thoroughness the authors have displayed, but also agrees with the reviewers that, when possible, higher quality references should be used when available and appropriate. If possible, try to limit the number of references to non-peer reviewed sources.

We are grateful to the Editor and the Experts for careful revision of our manuscript.

Reviewer comments to Author:

Reviewer: 1

Comments to the Author(s)

Please, check the text thoroughly, as there are some minor grammatical mistakes, although the text is readable and mostly clear.

Reviewer: 2

Comments to the Author(s)

Journal: Royal Society Open Science RSOS-191378

Reviewer's Comments

This manuscript has reported a complete and critical overview on several greener synthesis and fabrication of inorganic, organic and coordination compounds, materials, nanomaterials, hybrids, and nanocomposites. Details of greener techniques for synthesis of both well-known chemical compounds and completely new materials are also discussed. The manuscript is very exciting and topic is of scientific interest.

Authors have addressed and reviewed major physical methods as microwave heating, ultrasound-assisted and hydrothermal processes or ball milling utilized in the greener synthesis, as well as solventless and biosynthesis techniques. Then scaling up of green processes for profit are also explored. I think this manuscript is in very good shape and very important.

Overall, this paper is attractive from the standpoint of greener synthesis and fabrication research. The analysis of the topic is very significant and must be interesting to the readers of Royal Society Open Science. I strongly recommend it for publishing in Royal Society Open Science.

Reviewer: 3

Comments to the Author(s)

This review article by Kharisov and co-authors on greener synthesis of various compounds and materials is timely and useful. It covers several sub-topics. I recommend the publication after some changes and improvements. Hope that the following observations, comments and suggestions are useful to the authors and editor.

Title covers several topics. Different categories however are somewhat ill defined. What is the difference between each category such as "compounds, materials, hybrids, nanomaterials, and nanocomposites" listed in the title. For example, most of them can be described as materials or molecules and materials. Authors should think more carefully about this and use a better title.

Reply: we offer the following three titles:

Greener synthesis of compounds and materials: state of the art and modern trends.

or

Greener synthesis of chemical compounds and materials.

or

Greener synthesis of chemical compounds and materials: state of the art.

Methods of synthesis section is clear and nicely written. However, the type of items covered in sections 4-8 are less clear from reading first part of each section. It would be useful to describe what type of materials/molecules covered in each section in the first sentences of each sub-section. Also it is useful to direct the reader to any recent reviews on such topics in each section.

Reply: We inserted a short introduction (1-2 sentences) in each section, indicating main reviews and book chapters in every area.

Some editing at the editorial office needed to improve the language and presentation but overall it is good.

References should be improved significantly as they reflect on this review. There are inconsistencies, missing details, etc. Authors and journal office may want to check the list carefully -- See just a partial list below,

ref. 10, 17, 30, 48, 179, ..etc. have no titles or details apart from a web address.

Ref 73, 111, ...etc are incomplete

A few are from questionable and low-quality journals and non-peer reviewed items. Please use better options and improve

Reply: Improved. References completed, several new references added, some have been substituted.